# Relationship between uncertainty in the oil and stock markets before and after the shale gas revolution: Evidence from the OVX, VIX, and VKOSPI volatility indices

**Sun-Yong Choi** [1]*, **Changsoo Hong** [2]

**1** Department of Financial Mathematics, Gachon University, Gyeonggi, Republic of Korea, **2** NICE Pricing & Information Inc., Yeongdeungpo-gu, Seoul, Republic of Korea

* sunyongchoi@gachon.ac.kr

**Data Availability Statement:** The data are obtained from the CBOE and Korean stock exchange: http://www.cboe.com/; http://www.krx.co.kr/main/main.jsp.

## Abstract

We investigate the relationship between crude oil prices and stock markets. Unlike prior studies, we use implied volatility indices and evaluate the change in the relationship between the volatility indices through a sub-period analysis. Specifically, we examine the causal relationships among the crude oil, S&P 500 index, and KOSPI 200 index volatilities by using the autoregressive distributed lag (ARDL) bounds and the Toda–Yamamoto Granger causality tests. In addition, a BEKK-GARCH model is employed to enhance the robustness of the causality test results. These experiments indicate that the OVX and VIX show bi-directional causality in the period that includes the shale gas revolution and no causality in the period that does not. Further, the OVX Granger causes the VKOSPI in the former period, but there is no causality between them in the latter period. Finally, we find strong unidirectional causality from the VIX to the VKOSPI in both sub-periods. These results have important implications for the analysis of portfolio risk management and for assisting energy policymakers and traders in making effective decisions and investments, respectively.

## Introduction

Oil is one of the most important natural resources in the global economy. Many machines, such as cars, airplanes, and mechanical tools in factories, use oil as a power source. Moreover, numerous essential goods are manufactured from oil, such as plastic products and nylon clothing. Hence, oil is crucial to production activities and is becoming increasingly important for many countries.

Oil affects industrial development significantly and oil prices have naturally been the subject of global attention over the past several decades. A rise in crude oil prices increases the production cost of the manufacturing industry, reducing corporate profitability, which has a negative effect on stock prices (e.g. [1]). This is because increased crude oil price volatility can negatively affect economic growth, causing greater economic uncertainty. Empirical test

**Funding:** The work of S.-Y. Choi was supported by the Gachon University research fund of 2018 (GCU-2018-0295) and by the National Research Foundation of Korea (NRF) grant funded by the Korea government (MSIT) (No.2019R1G1A1010278). The funders had no role in study design, data collection and analysis, decision to publish, or preparation of the manuscript. The funder (NICE Pricing & Information) provided support in the form of salaries for author C. Hong, but did not have any additional role in the study design, data collection and analysis, decision to publish, or preparation of the manuscript. The specific roles of these authors are articulated in the 'author contributions' section.

results indicating that crude oil prices and economic activity are very much related are already seen in many studies ([2], [3], [4], [5], [6], [7], [8], [9], [10], [11], [12]).

Because oil is so important, every country is affected by changes in the oil market. For example, during oil crises (e.g., the 1973 oil crisis), rising oil prices have a dramatic effect on oil-exporting nations, which then accumulate vast wealth. On the contrary, for oil-importing countries, oil price increases lead to significant slowdowns in economic growth.

Two consecutive oil shocks in the early and late 1970s resulted in many studies that investigate the effect of oil price changes on the economic or financial environment. A number of them investigate how oil shocks affect macroeconomic variables such as GDP (gross domestic product), inflation, exchange rates, and government expenditure ([13], [14], [15], [16], [17], [18]). Moreover, a few studies examine the relationship between oil prices and exchange rates intensively. For instance, some use the nominal and real dollar exchange rates to examine the relationship ([19], [20]), while others analyze the effect of oil shocks on exchange rates for oil-importing and -exporting countries ([21], [22], [23]).

In addition, some past studies focus on oil price volatility and its relationship with other economic variables. [24] examine the volatility of crude oil prices and find stylized facts and permanent and asymmetric effects. [25] uses volatility models that allow for two structural breaks and finds evidence of persistence and leverage effects in oil price volatility. [26] forecasts oil price volatility using a hybrid model combining artificial neural networks and the GARCH model. In addition, some works examine the volatility transmission between oil and other assets such as agricultural commodities prices ([27], [28], [29], [30]).

Oil price volatility measures the uncertainty of oil prices in the market. High volatility means large fluctuations in oil prices, which is undesirable for both oil-exporting and -importing countries. The greater the uncertainty in oil prices, the higher the cost of managing this resource. Therefore, observing oil price volatility and taking its expected changes into account are essential for managing risk.

Most research still uses crude oil and stock prices. However, the volatility indices are a better suitable barometer of the fragility of the markets and the economy. Therefore, the aim of this work is to investigate the relationship among the volatility indices, to derive important implications for the analysis of portfolio risk management. Furthermore, since the introduction of volatility derivatives (e.g., Chicago Board Options Exchange (CBOE) volatility index (VIX) futures, options, and exchange-traded products), the trading volume has been increasing because they can be used as a risk-hedging strategy against stock market downturns (e.g. [31]). Accordingly, investigation of the relationship between volatility indices can give necessary insight into suggestions for the pricing of volatility derivatives.

We use the CBOE crude oil volatility index (OVX) to proxy for oil price volatility. The OVX is a market estimate of the expected 30-day volatility in crude oil prices and is thus regarded as a measure of oil market uncertainty. The OVX is calculated by applying the VIX methodology to the United States Oil Fund options. The United States Oil Fund is an exchange-traded security designed to track daily price movements in West Texas Intermediate (WTI) light, sweet crude oil.

As measures of stock market volatility, we adopt the VIX and VKOSPI volatility indices, calculated from S&P 500 index options and the KOSPI 200 index, respectively. The VIX, created by the CBOE, measures the expected 30-day volatility in the U.S. stock market. Notably, the VKOSPI is the first volatility index for a domestic Asian stock market. It represents the expected volatility of the next 30 days for the KOSPI 200 index option.

Because South Korea has close political, economic, and social ties with the United States, it is worthwhile examining the relationship between the volatility indices of the U.S. and South Korean stock markets. However, South Korea's GDP is only ranked 12th in the world,

according to the World Bank (https://datacatalog.worldbank.org/dataset/gdp-ranking). In addition, according to the US Energy Information Administration, it is among the 10 largest oil consumers and depends entirely on oil imports to meet its needs. Consequently, the South Korean economy is sensitive to variations in oil prices.

Methodologically, we adopt the autoregressive distributed lag (ARDL) bounds test for cointegration as well as the Toda–Yamamoto (TY) Granger causality test developed in [32]. To enhance the robustness of the tests, we employ a multivariate model introduced by [33] called the BEKK model. Furthermore, these tests are explored using a sub-period analysis to examine whether their relationship is constant over time, which would provide insight into the dynamic nature of the interactions between the volatility indices. The other reason we proceed with the sub-period analysis is because we want to analyze how shale gas, an alternative to crude oil, affects the relationship between stock markets and the oil market. Several studies investigate the effect of the shale gas revolution on the oil market.

There are two aspects of this study that differ from previous studies. The first is our use of volatility indices to identify the relationship between crude oil and the stock market. Although previous empirical studies find causal relationships between oil prices and stock indices, research on the causality between implied volatility indices is scarce. To bridge this gap, we adopt the OVX, VIX, and VKOSPI to measure the implied volatility in oil prices, the S&P 500, and the KOSPI 200, respectively. Second, whereas previous studies focus mainly on the relationship between crude oil and the stock market, we focus on the change in that relationship over time.

We obtain three main contributions from these differences. The first is the investigation of the relationship between future expectations for each market—the crude oil, U.S., and South Korean stock markets. In particular, the volatility index represents the future risk measure of market participants. Therefore, we can investigate the relationship between the risk measures implied by crude oil, the S&P 500 index, and the KOSPI 200 index by using the volatility indices. The second is the examination of the causality between the OVX and VKOSPI and between the VIX and VKOSPI. To the best of our knowledge, this study is the first to investigate the relationship between the OVX and VKOSPI. Based on their relationship, policymakers can propose laws and policies for oil-importing countries to manage market risk. As mentioned above, South Korea and the United States have a close economic relationship; hence, it is reasonable to explore the causality between them owing to the uncertainty in their stock markets. The third major contribution concerns the change in the relationship between the volatility indices as revealed through a sub-period analysis. Based on the empirical results of the sub-period analysis, we conclude that one of the factors causing the change in the relationship is the increased production of shale gas. Detailed discussions on this will be covered in Section 6.

The rest of the paper is organized as follows. Section 2 presents a review of the literature on volatility indices. Section 3 describes the data and methodology used in this study. Section 4 provides the results of the empirical analysis for the full sample period. In Section 5, we show the sub-period analysis. Section 6 presents a discussion of the results in terms of shale gas and risk management. Lastly, we provide concluding remarks in Section 7.

## Literature review

There is a vast body of literature on the implied volatility indices. In this section, we would like to divide the research into that on the VIX, OVX, and VKOSPI according to the content.

First, because the VIX is obtained from the S&P 500 index, many studies have focused on the relationship between the VIX (implied volatility) and S&P 500 index (underlying asset).

For example, [34] investigates the relationship between the implied volatility and underlying stock index for both the S&P 100 and Nasdaq 100 indices. According to [34], there is a negative relationship between changes in the VIX and the underlying stock index. Other studies also obtain similar results ([35], [36], [37], [38]), while some investigate the behavior of the VIX and VIX futures markets ([39], [40], [41], [42], [43]). Furthermore, the VIX has also been used to explain market change, because it represents the market's expectations regarding future uncertainties. For example, many studies investigate the effects of macroeconomic events on the VIX. [44] show how the VIX behaves around the time of monetary policy announcements. [45] find that good and bad news affect the VIX asymmetrically.

Since the launch of the VIX, the huge demand for volatility risk hedging has led to increasing trade in volatility derivatives ([46], [47], [48]). Furthermore, attempts are being made to model VIX and VIX derivatives such as VIX options ([49], [50], [51], [52], [53], [54], [55]).

Recently, some studies have examined VIX term structures. [56] investigate the term structure of VIX and present a VIX formula under the general jump-diffusion model. [57] shows that two factors (Level and Slope) can explain the dynamics of the VIX term structure effectively. [58] analyzes how principal components of the S&P 500 index affect the VIX term structure.

Similarly, much research examines the relationship between the OVX and crude oil price ([59], [60, 61], [62]). In addition, a number of studies look into the relationship between various market variables. Much of the research is done on the relationship between the stock markets. [63] study the relationship between the OVX and alternative energy sector equity. [64] explore the effect of OVX shocks on the Chinese stock market index. [65] investigate whether the OVX affects the Middle Eastern and African stock markets. [66] examine whether the OVX improves the directional predictability of the implied volatility index for some stock markets(France, Germany, India, Japan, Mexico, the Netherlands, Russia, South Africa, Sweden, Switzerland, the United Kingdom, and the United States).

Many studies look into the relationship with economic variables without stock. Because crude oil is classified as a commodity, many studies look at its relationship with other commodities. [67] investigates the cross-market uncertainty transmission implied by the OVX and other volatility indices (VIX, EVZ, and GVZ). The EVZ and GVZ are the euro/dollar exchange rate and the gold price volatility indices, respectively. Adopting a similar approach, [68] studies the implied volatility transmission across commodity, equity, foreign exchange, and Treasury bond markets by using EVZ, GVZ, OVX, VIX, and VXTYN. The VXTYN is the Treasury note futures price volatility index. [69] investigate the predictive power of OVX in explaining the return structure of the precious metal (gold and silver) markets. [70] investigates the dependence structure between OVX, WIV (wheat volatility index), and CIV(corn volatility index) during bear, normal, and bull markets. [71] examine the effects of OVX on the returns and volatilities of Chinese commodities (petrochemicals, agricultural commodities, and metals). By contrast, credit-related studies have been also conducted. [72] examine the directional predictability from the OVX to the sovereign CDS spreads of oil-exporting countries. [73] investigate the dependence between OVX and BRICS sovereign CDS spreads from July 2009 to March 2017. [74] investigate the dynamic spillover of the OVX and volatilities on sovereign credit default swap (CDS) spreads of ten oil-exporting countries.

There are several studies that use the VKOSPI to investigate the characteristics of the South Korean market ([75], [76], [77]). However, few studies explore the relationship between foreign markets and South Korean markets using the VKOSPI. According to [77], U.S. market factors are more important than South Korean factors in explaining VKOSPI dynamics. Tables 1–3 provide a summary of each study included in the review.

**Table 1. The first literature review summary.**

| Study | Main Data | Relevant findings |
|---|---|---|
| [35] | S&P 100, VIX | A large negative contemporaneous correlation exists between VIX changes and S&P 100 index returns |
| [36] | S&P 100, VIX | VIX has acted reliably as a fear gauge. |
| [34] | S&P 100, VIX NASDAQ100, VXN | There is a strong negative relationship between contemporaneous changes in implied volatility indexes and the underlying stock indexes |
| [37] | S&P500, VIX | A strongly negative contemporaneous correlation exists between the VIX and SPX returns |
| [39] | VIX | Develops a VIX model to price VIX futures |
| [46] | S&P 500, VIX | Examines the benefits of adding VIX to the S&P 500 stock portfolio for reducing risk |
| [40] | VIX | Examines the driving of the dynamics of implied volatility indices in continuous time. |
| [44] | VIX | VIX reacts to U.S. monetary policy announcements |
| [51] | VIX, VIX option | Develops a dynamic model for the joint evolution of the VIX spot value and the S&P500 index to evaluate VIX futures and options. |
| [41] | S&P 500, VIX | VIX does reflect past jump activity in the S&P 500. |
| [38] | S&P 500, VIX | Investigates the inverse relation between movements in the VIX and movements in the S&P 500 |
| [50] | VIX | Provides closed-form valuation models for European options written on the spot and forward VIX, respectively. |
| [42] | VIX future | Develops a general model to price VIX futures contracts |
| [52] | VIX and SPX options | Examines the pricing performance of VIX option models |
| [56] | 1,3,6,9,12,15 months VIX | Presents the VIX formula under the general jump-diffusion model. |
| [43] | VIX futures | Develops a term structure model for VIX futures |
| [53] | VIX futures | Presents a general theory and a unifying framework for understanding arbitrage pricing theory in VIX options |

To the best of our knowledge, there is no study investigating the links between the VIX, OVX, and VKOSPI. Instead, many investigate the VIX and OVX ([67], [68], [78], [79], [80]). Among them, [78], [79], [80] study the link between the VIX and OVX. We contribute to the finance literature by uncovering the relationship between the VIX, OVX, and VKOSPI under different market conditions.

## Data and methodology

### Data

The volatility time series we use in this study consists of three indices: the OVX, VIX, and VKOSPI. The data are obtained from the CBOE and South Korean stock exchange. The sample period runs from January 2, 2009 to December 28, 2018, yielding 2354 daily observations.

Table 4 shows the descriptive statistics for each volatility index. Panel A presents the results for the log prices; whereas, Panel B does the same for the differenced series. The average OVX is the highest and all the series show positive skewness. Furthermore, all the first-differenced indices have a leptokurtic distribution with asymmetric tails as supported by the corresponding kurtosis results. The Jarque–Bera test implies that the normality hypothesis is rejected. Moreover, the Lagrange multiplier test indicates the existence of the autoregressive conditional heteroscedastic(ARCH) effect in the average log returns of all the volatility indices.

Table 5 presents the Pearson correlation coefficients among the volatility indices, showing significantly positive correlations, which indicate that the expected changes in oil and stock

**Table 2. The second literature review summary.**

| Study | Main Data | Relevant findings |
|---|---|---|
| [54] | | Presents an analytical exact solution for the price of VIX options under stochastic volatility model with jumps |
| [45] | SPX option, VIX | Good and bad announcements change implied volatility slope and VIX. |
| [55] | S&P 500, VIX | Develops a regime-switching Heston model |
| [49] | | Demonstrates a 3/2 model for the pricing of equity and VIX derivatives |
| [47] | S&P 500, VIX futures | Proposes a methodology using VIX futures as an investment asset |
| [48] | S&P 500, VIX ETPs(exchange traded products) | Provides an analysis of VIX ETPs with a focus on hedging |
| [57] | S&P500 variance swap, VIX | Changes in the VIX term structure convey information about variance risk premia rather than expected changes in the VIX |
| [58] | 9 days, 1,3,6,12 months VIX | Identifies the principal factors affecting the change in volatility term structure |
| [67] | VIX, OVX, EVZ, GVZ | There is no strong long-run equilibrium relationship between the OVX and other volatility indices |
| [68] | EVZ, GVZ, OVX, VIX, VXTYN | Volatility contagion across U.S. equity and non-equity markets |
| [59] | crude oil price, OVX | A negative and asymmetric contemporaneous relationship between OVX changes and crude oil price returns |
| [60] | VIX, USO crude oil ETF | A negative relationship between the contemporaneous oil VIX and USO ETF oil returns |
| [61] | crude oil price, OVX | The OVX contains information regarding the future realized volatility of crude oil returns |
| [62] | WTI, Brent, OVX | The OVX has predictive ability for spot volatility of WTI and Brent oil returns |
| [63] | WTI, OVX, Clean Energy Index | OVX improves the volatility forecasts for the clean energy equity market |
| [64] | Shanghai Composite Index, WTI, OVX | Oil price shocks positively affect Chinese stock returns |
| [70] | OVX,WIV, CIV | Evidence of asymmetric tail dependence between the pair of cereals as well as between oil |

markets have the same movement over the sample period. In addition, the highest correlation is observed between the volatility indices of the U.S. and South Korean stock markets.

Fig 1 illustrates the volatility indices used in this study for the whole sample period. Several sharp fluctuations occur due to drastic changes in the political, business, or economic environment. For example, the European debt crisis (2010), Black Monday (2011), and the Libyan War (2011) all cause large fluctuations in the oil and stock markets. In particular, certain spikes are observed in the OVX series during 2015 and 2016. These spikes are provoked by the strong U.S. dollar, OPEC's control, and Iran's nuclear deal.

## ARDL bounds tests

The cointegration tests proposed by [81], [82], and [83] have been used in many empirical studies to investigate the long-run relationship of economic variables. However, the use of these approaches is limited. For example, these methods can be applied to those series that have a unique order of integration. The ARDL bounds test proposed by [84] and [85] is a popular method because it has certain advantages over traditional cointegration methods.

First, it does not need all the variables in the model to be integrated of the same order. Second, the approach is relatively more efficient in the case of small and finite sample data sizes

**Table 3. The third literature review summary.**

| Study | Main Data | Relevant findings |
|---|---|---|
| [65] | OVX, Stock market indices | Oil market uncertainty has substantial effects on the realized volatility of most Middle Eastern and African stock markets |
| [72] | WTI, sovereign CDS spreads, OVX | A significant directional predictability from oil uncertainty to the CDS spreads for most oil-exporting countries. |
| [73] | OVX, CDS spreads BRICS countries | Low (high) volatility of the oil market predicts low (high) sovereign risk |
| [66] | OVX, implied volatility indices | The ability of crude oil to predict stock market conditions using implied volatility data and a cross-quantilogram approach |
| [69] | OVX, WTI, S& P GSCI data | A significant price spillover running from the oil market to the industrial metal sector |
| [74] | OVX, WTI, Sovereign CDS | Study the dynamic spillover from crude oil prices to sovereign CDS spreads |
| [71] | WTI, OVX, industrial sectors of China | China's commodity returns respond negatively to OVX shock |
| [75] | KOSPI 200, VKOSPI | A strong asymmetric and negative relationship between KOSPI 200 returns and the VKOSPI |
| [76] | VKOSPI, Common stocks listed on the KOSPI | Future returns on large stocks are higher than those on small stocks on days |
| [77] | S&P 500, VKOSPI | U.S. market factors are more significant than domestic (Korean) factors in explaining VKOSPI dynamics |
| [78] | OVX, Implied volatility indices | The connectedness between oil and equity is established by the bi-directional information spillovers between the two markets |
| [79] | S&P 500, OVX | A strong co-movement between the volatilities of the equity and oil markets |
| [80] | VIX, VXXLE(U.S. energy sector equity VIX), OVX | There is a long-run relationship between oil and stock market implied volatility indexes |

([84] and [86]). Third, applying the ARDL technique, we obtain unbiased estimates of the long-term model ([87]).

We use this approach to analyze the dynamic interactions between oil prices and stock markets. The ARDL bounds test assumes that the variables are integrated of order zero ($I(0)$) or

**Table 4. Summary statistics for the VIX and VKOSPI.** [‡] and [†] indicate the rejection of the null hypothesis at the 1% and 5% significance levels, respectively.

| Index | log (OVX) | log (VIX) | log (VKOSPI) |
|---|---|---|---|
| Panel A: Log price | | | |
| Mean | 3.499883 | 2.850168 | 2.822976 |
| Standard deviation | 0.334504 | 0.352381 | 0.332613 |
| Skewness | 0.102328 | 0.771809 | 1.104779 |
| Kurtosis | -0.161943 | 0.204169 | 1.063548 |
| Jarque–Bera Test | 6.7344[†] | 242.65[‡] | 602.16 [‡] |
| Lagrange multiplier test | 2292.1[‡] | 2234.5[‡] | 2285.2[‡] |
| Panel B: First difference | | | |
| Mean | -0.000209 | -0.000135 | -0.000386 |
| Standard deviation | 0.050136 | 0.077759 | 0.055497 |
| Skewness | 0.762307 | 1.048369 | 0.967227 |
| Kurtosis | 9.953206 | 7.379172 | 6.524589 |
| Jarque–Bera Test | 10148[‡] | 5890.8[‡] | 4636.3[‡] |

**Table 5. Correlation coefficients.** The sample period ranges from 2009 to 2018.

| Panel A: Log price | log (OVX) | log (VIX) | log (VKOSPI) |
|---|---|---|---|
| log (OVX) | 1.0000 | 0.6313 | 0.5529 |
| log (VIX) | 0.6313 | 1.0000 | 0.8989 |
| log (VKOSPI) | 0.5529 | 0.8989 | 1.0000 |
| Panel B: First difference | Δlog (OVX) | Δlog (VIX) | Δlog (VKOSPI) |
| Δlog (OVX) | 1.0000 | 0.4438 | 0.1946 |
| Δlog (VIX) | 0.4438 | 1.0000 | 0.1955 |
| Δlog (VKOSPI) | 0.1946 | 0.1955 | 1.0000 |

integrated of order one ($I(1)$). Therefore, we determine the order of integration of all the variables before applying this test.

The *F*-test (or the Wald test) can be performed in order to check the significance of the lagged coefficient in the unrestricted correction model. The critical values comprise the upper and lower bound values following [85]. If the calculated *F*-statistic is above the upper critical value, we can conclude that a long-term relationship exists. If the calculated *F*-statistic is below the lower critical value, we cannot reject the null hypothesis of no cointegration. If it lies between the upper and lower bound critical values, then the decision is inconclusive.

The ARDL bounds test can be applied in two steps. The first is determining the existence of the long-run relationship between the variables using the *F*-statistic. The second step is choosing the appropriate lag order for the ARDL model and estimating the long-run estimates of the selected ARDL model. If a long-run relationship exists between the underlying variables, the ARDL approach to cointegration can be applied. Using model order selection criteria such as the Akaike information criterion (AIC) and Schwarz criterion (SC), we determine the optimal lag length for the ARDL model. Under the best model, the estimates become the long-run coefficients.

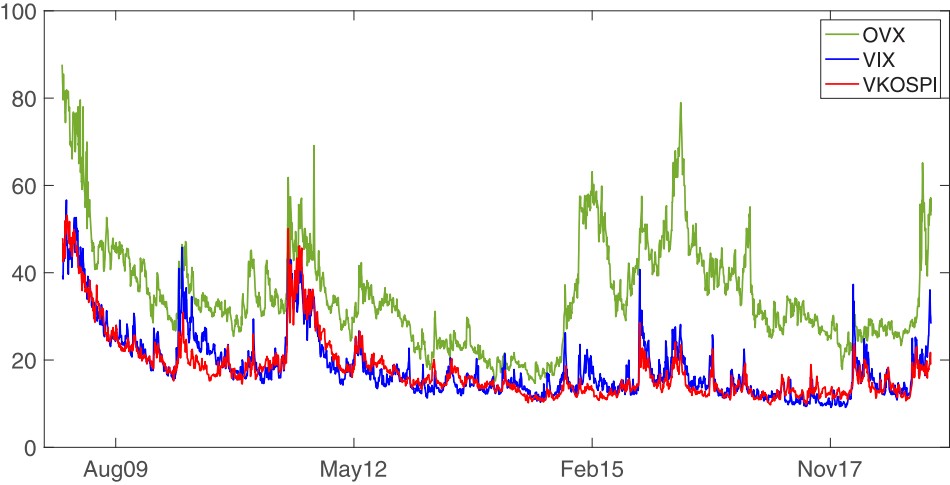

**Fig 1. Volatility indices for the sample period from January 2009 to December 2018.**

The ARDL model used in this study is expressed as follows:

$$
\begin{aligned}
\Delta \log(OVX)_t \;=\;& \omega_1 + \sum_{i=1}^{p} \alpha_{1,i}\Delta\log(OVX)_{t-i} + \sum_{i=0}^{q} \beta_{1,i}\Delta\log(VIX)_{t-i} \\
& + \sum_{i=0}^{r} \gamma_{1,i}\Delta\log(VKOSPI)_{t-i} + a_1 \log(OVX)_{t-1} \\
& + b_1 \log(VIX)_{t-1} + c_1 \log(VKOSPI)_{t-1} + \epsilon_{1,t},
\end{aligned}
\tag{1}
$$

$$
\begin{aligned}
\Delta \log(VIX)_t \;=\;& \omega_2 + \sum_{i=0}^{p} \alpha_{2,i}\Delta\log(OVX)_{t-i} + \sum_{i=1}^{q} \beta_{2,i}\Delta\log(VIX)_{t-i} \\
& + \sum_{i=0}^{r} \gamma_{2,i}\Delta\log(VKOSPI)_{t-i} + a_2 \log(OVX)_{t-1} \\
& + b_2 \log(VIX)_{t-1} + c_2 \log(VKOSPI)_{t-1} + \epsilon_{2,t},
\end{aligned}
\tag{2}
$$

$$
\begin{aligned}
\Delta \log(VKOSPI)_t \;=\;& \omega_3 + \sum_{i=0}^{p} \alpha_{3,i}\Delta\log(OVX)_{t-i} + \sum_{i=0}^{q} \beta_{3,i}\Delta\log(VIX)_{t-i} \\
& + \sum_{i=1}^{r} \gamma_{3,i}\Delta\log(VKOSPI)_{t-i} + a_3 \log(OVX)_{t-1} \\
& + b_3 \log(VIX)_{t-1} + c_3 \log(VKOSPI)_{t-1} + \epsilon_{3,t},
\end{aligned}
\tag{3}
$$

where $\Delta$ and $\epsilon_{i,t}(i = 1, 2, 3)$ are the first-difference operator and white noise error terms. The null hypothesis of no cointegration among the volatility indices in (1) is

$$
H_0 : a_1 = b_1 = c_1 = 0
$$

against the alternative hypothesis:

$$
H_1 : a_1 \neq b_1 \neq c_1 \neq 0.
$$

In model (1), we denote the $F$-statistic of the test by $F_{OVX}(OVX \mid VKOSPI, VIX)$. In models (2) and (3), we denote the $F$-statistic as $F_{VIX}(VIX \mid OVX, VKOSPI)$ and $F_{VKOSPI}(VKOSPI \mid OVX, VIX)$, respectively.

## TY granger causality tests

[88] proposes a test of the causal relationship between two variables, known as Granger causality. A time series ($X$) is said to Granger cause another time series ($Y$) if the prediction error of the current $Y$ declines by using the past values of $X$ in addition to the past values of $Y$. In the test, the two variables are expressed by simple vector autoregression (VAR). The Granger causality test is easy to implement and can be applied in many types of empirical studies. Nonetheless, it also has some drawbacks. According to [89], first, the Granger causality test for inferring the leads and lags among integrated variables can provide spurious regression results. Second, it does not consider the effect of the number of lags even though this can affect the performance of a causality test. In other words, the results of the Granger causality test depend on the number of lags. Moreover, [90] insist that Granger causality can lead to drawing wrong conclusions because of the dependence of the parameters.

[32] introduce a simple procedure involving the estimation of an augmented VAR. As [91] states, the TY approach uses a modified Wald test to restrict the parameters of the VAR ($m$).

The VAR system is then augmented by the maximum order of integration (*dmax*). The VAR (*m* + *dmax*) is estimated without the coefficients of the last lagged *dmax* vector. The Wald statistic asymptotically follows the chi-square distribution with degrees of freedom equal to the number of the excluded lagged variables.

Several other methodologies have been developed since the TY causality test was introduced. In view of the TY causality test being limited to finding linear cause-effect relationship, nonlinear version Granger causality tests have been developed(e.g. [92] and [93]). We can detect nonlinear interactions between the variables using these tests. By contrast, [94] proposes an asymmetric causality test for the existence and direction of causality. Thus, the causality between positive and negative shocks of variables can be determined in the method. In addition, [95] introduces a bootstrap panel causality method in order to account for both cross-sectional dependence and slope heterogeneity ([96]). The approach is widely used to test for causality in a panel framework in many empirical studies.

The TY Granger causality test has several advantages over other methods. First, it can provide a valid result regardless of whether a series is $I(0)$, $I(1)$, or $I(2)$, not cointegrated, or cointegrated of any arbitrary order. Second, the TY test avoids the bias associated with unit root and cointegration tests ([97], [98]) as it does not require pre-testing of the cointegrating properties of the system. Third, we can explore the causality between variables with a possibly integrated and cointegrated system using the augmented VAR model in the TY test because the long-run information of the system in the general VAR model often disappears in the mandatory process of first differencing and pre-whitening ([98], [91]). Therefore, we adopt TY causality testing in this study. Furthermore, many recent studies adopt the approach of identifying causality using the TY causality test ([99], [100], [101], [102], [103]).

We use the following three-variable VAR model:

$$
\begin{aligned}
X_t = {} & \omega_1 + \sum_{i=1}^{m}\theta_{1,i}X_{t-i} + \sum_{i=m+1}^{m+dmax}\theta_{1,i}X_{t-i} + \sum_{i=1}^{m}\delta_{1,i}Y_{t-i} \\
& + \sum_{i=m}^{m+dmax}\delta_{1,i}Y_{t-i} + \sum_{i=1}^{m}\gamma_{1,i}Z_{t-i} + \sum_{i=m}^{m+dmax}\gamma_{1,i}Z_{t-i} + \epsilon_{1,t},
\end{aligned}
\tag{4}
$$

$$
\begin{aligned}
Y_t = {} & \omega_2 + \sum_{i=1}^{m}\theta_{2,i}X_{t-i} + \sum_{i=m+1}^{m+dmax}\theta_{2,i}X_{t-i} + \sum_{i=1}^{m}\delta_{2,i}Y_{t-i} \\
& + \sum_{i=m}^{m+dmax}\delta_{2,i}Y_{t-i} + \sum_{i=1}^{m}\gamma_{2,i}Z_{t-i} + \sum_{i=m}^{m+dmax}\gamma_{2,i}Z_{t-i} + \epsilon_{2,t},
\end{aligned}
\tag{5}
$$

$$
\begin{aligned}
Z_t = {} & \omega_3 + \sum_{i=1}^{m}\theta_{3,i}X_{t-i} + \sum_{i=m+1}^{m+dmax}\theta_{3,i}X_{t-i} + \sum_{i=1}^{m}\delta_{3,i}Y_{t-i} \\
& + \sum_{i=m}^{m+dmax}\delta_{3,i}Y_{t-i} + \sum_{i=1}^{m}\gamma_{3,i}Z_{t-i} + \sum_{i=m}^{m+dmax}\gamma_{3,i}Z_{t-i} + \epsilon_{3,t},
\end{aligned}
\tag{6}
$$

where $X$ = log (OVX), $Y$ = log (VIX), and $Z$ = log (VKOSPI). The $\omega$'s, $\theta$'s, $\delta$'s, and $\gamma$'s are the parameters of the model. The $\epsilon$'s are the white noise error terms and *dmax* is the maximum order of integration.

To test the hypothesis of "no Granger causality from $Y$ to $X$," we use the null hypothesis

$$
H_0 : \delta_{1,1} = \delta_{1,2} = \ldots = \delta_{1,m} = 0,
$$

against the alternative hypothesis

$$H_1 : \delta_{1,1} \neq \delta_{1,2} \neq \ldots \neq \delta_{1,m} \neq 0.$$

We need to determine the optimal lag length ($m$) and maximum order of integration ($dmax$) to implement the TY Granger causality test and we use the AIC, SC, final prediction error (FPE), and Hannan–Quinn (HQ) information criteria to choose the appropriate lag order ($m$) of VAR models (4)–(6). Furthermore, we use several unit root tests to find the maximum order of integration ($dmax$), as detailed in the following section.

## Multivariate GARCH model

To enhance the robustness of the ARDL and TY causality tests, we use a multivariate model introduced by [33] called the BEKK model. This model has been used extensively to examine shock and volatility spillover effects ([104], [105], [106], [28], [107], [108], [109]).

In light of the evidence of the ARCH effects reported in Table 1, we exploit the econometric specifications used to analyze the shock and volatility dynamics of the volatility indices, which consist of the conditional mean equation and covariance models.

We first introduce the conditional mean equation, as defined by:

$$r_t = \bar{\mu} + \epsilon_t \quad \epsilon_t = \sqrt{H_t} v_t \tag{7}$$

where $r_t$ is a $n \times 1$ vector of average log-returns for $n$ different sectors, $\bar{\mu}$ is the $n \times 1$ mean of the returns, and $\epsilon_t$ is an $n \times 1$ vector of zero-mean error terms with conditional covariance matrix $H_t$. $v_t$ is an $n \times 1$ vector of standardized residuals.

For the conditional variance–covariance equations, we employ the BEKK–GARCH(1,1) model. This model allows us to describe shock and volatility spillover effects. The conditional covariance matrix of the BEKK model, $H_t$, is expressed as:

$$H_t = C'C + A'\epsilon_{t-1}\epsilon'_{t-1}A + B'H_{t-1}B, \tag{8}$$

where $H_t$ is the $n \times n$ covariance matrix, $A$, $B$, and $C$ are $n \times n$ matrices, and $C$ is an upper triangular matrix. Matrices $A$ and $B$ are ARCH and GARCH parameters, respectively. Furthermore, $\epsilon_{t-1}$ is the $n \times 1$ vector of error terms in (7).

In this study, we use a three-variate($n = 3$) BEKK model denoted by BEKK–GARCH(1,1). That is, the return vector $r_t$ is a vector of $(X_t - X_{t-1}, Y_t - Y_{t-1}, Z_t - Z_{t-1})^T$, where $X = \log$ (OVX), $Y = \log$ (VIX), and $Z = \log$ (VKOSPI). In matrix form, it can be written as:

$$
\begin{pmatrix} h_{11,t} & h_{12,t} & h_{13,t} \\ h_{21,t} & h_{22,t} & h_{23,t} \\ h_{31,t} & h_{32,t} & h_{33,t} \end{pmatrix} = C'C + A' \begin{pmatrix} \epsilon_{1,t-1}^2 & \epsilon_{1,t-1}\epsilon_{2,t-1} & \epsilon_{1,t-1}\epsilon_{3,t-1} \\ \epsilon_{2,t-1}\epsilon_{1,t-1} & \epsilon_{2,t-1}^2 & \epsilon_{2,t-1}\epsilon_{3,t-1} \\ \epsilon_{3,t-1}\epsilon_{1,t-1} & \epsilon_{3,t-1}\epsilon_{2,t-1} & \epsilon_{3,t-1}^3 \end{pmatrix} A
$$
$$
+ B' \begin{pmatrix} h_{11,t-1} & h_{12,t-1} & h_{13,t-1} \\ h_{21,t-1} & h_{22,t-1} & h_{23,t-1} \\ h_{31,t-1} & h_{32,t-1} & h_{33,t-1} \end{pmatrix}, \tag{9}
$$

where

$$C = \begin{pmatrix} c_{11} & c_{12} & c_{13} \\ 0 & c_{22} & c_{23} \\ 0 & 0 & c_{33} \end{pmatrix}, A = \begin{pmatrix} a_{11} & a_{12} & a_{13} \\ a_{21} & a_{22} & a_{23} \\ a_{31} & a_{32} & a_{33} \end{pmatrix}, B = \begin{pmatrix} b_{11} & b_{12} & b_{13} \\ b_{21} & b_{22} & b_{23} \\ b_{31} & b_{32} & b_{33} \end{pmatrix}.$$

According to the matrix equation of the BEKK–GARCH(1,1) model (9), the diagonal elements($a_{ii}$ and $b_{ii}$) in matrices $A$ and $B$ represent a sector's own ARCH and GARCH effects, respectively. By contrast, the off-diagonal elements of matrices $A$ and $B$($a_{ij}$ and $b_{ij}$, $i \neq j$) capture the market shock and volatility spillovers, respectively. In detail, the off-diagonal elements $a_{ij}$ show the effect of index $i$'s change on index $j$'s volatility and the off-diagonal elements $b_{ij}$ measure the effects of past volatility of index $i$ on index $j$'s conditional variance.

## Empirical results

### Unit root tests and bounds tests

Before we proceed with the ARDL bounds test, we test for the stationarity of the volatility indices. Because the ARDL bounds test assumes that the variables are $I(0)$ or $I(1)$, it is necessary to determine their order of integration to avoid spurious results. We adopt the augmented Dickey–Fuller (ADF), Phillips–Perron (PP), and Kwiatkowski–Phillips–Schmidt–Shin (KPSS) tests to check the presence of a unit root.

Table 6 presents the results for the ADF, PP, and KPSS unit root tests for the three return series. The results of the stationarity tests show that all log prices are non-stationary. However, the unit roots for all the first differences of log prices can be rejected. In other words, the first-differenced series for the three stock indices are stationary for the entire sample period (from 2009 to 2018). Therefore, we can conclude that all the variables are $I(1)$.

To proceed with the ARDL bounds test, it is necessary to determine the lag structure of the examined variables under models (1)–(3). We adopt the AIC to choose the appropriate lag structure. Table 4 reports the selection of the optimal lag.

Table 7 presents the calculated $F$-statistics. Their values for (1) are $F_{OVX}$(OVX | VIX, VKOSPI) = 7.64; for (2) are $F_{VIX}$(VIX | OVX, VKOSPI) = 25.29; and for (3) are $F_{VKOSPI}$(V-KOSPI | OVX, VIX) = 18.81. These results show that there are long-term relationships among the volatility indices because their calculated $F$-statistics are above the upper bound critical value of $I(1)$ = 6.36 at the 1% significance level. In other words, the null hypothesis of no cointegration among the variables in Eqs (1)–(3) is rejected. The presence of long-term relationships suggests the existence of causal relationships among these volatility indices.

**Table 6. The results of the ADF, PP, and KPSS unit root tests on data in log price and first-differenced forms.**

| Index | ADF | PP | KPSS |
|---|---|---|---|
| Panel A: Log price | | | |
| log (OVX) | -3.9949[‡] | -3.6754[‡] | 2.6894[‡] |
| log (VIX) | -6.7962[‡] | -4.8219[‡] | 12.163[‡] |
| log (VKOSPI) | -5.7964[‡] | -4.1747[‡] | 14.134[‡] |
| Panel B: First difference | | | |
| Δlog (OVX) | -53.878[‡] | -54.6988[‡] | 0.1529 |
| Δlog (VIX) | -52.885[‡] | -54.3461[‡] | 0.0576 |
| Δlog (VKOSPI) | -50.126[‡] | -51.2814[‡] | 0.0828 |

**Table 7. Results of the ARDL bounds tests.**

| Model | Optimal lag $(p, q, r)$ | $F$-statistic | Decision |
|---|---|---|---|
| $F_{OVX}$(OVX | VIX, VKOSPI) | (1,1,1) | 7.64 | Long-term relationship exists |
| $F_{VIX}$(VIX | OVX, VKOSPI) | (1,1,1) | 25.29 | Long-term relationship exists |
| $F_{VKOSPI}$(VKOSPI | OVX, VIX) | (1,2,1) | 18.81 | Long-term relationship exists |
| Critical values | 1% | 5% | 10% |
| Lower bounds $I(0)$ | 5.15 | 3.79 | 3.17 |
| Upper bounds $I(1)$ | 6.36 | 4.85 | 4.14 |

## TY granger causality tests

We implement the TY Granger causality test to investigate the direction of causality among the volatility indices. As stated in the subsection TY Granger causality tests, we determine the lag length ($m$) and maximum order of integration ($dmax$). To determine the optimal lag length of the VAR system, we use four information criteria (AIC, HQ, SC, and FPE). Table 8 displays the results of selecting the optimal lag ($m = 7$) of VAR models (4)–(6). In addition, based on the unit root test, the maximum order of integration for the volatility indices is one ($dmax = 1$).

Table 9 presents the results of the VAR model. First, these results suggest that both the null hypothesis of "no Granger causality from OVX to VIX" and "no Granger causality from total VIX to OVX" can be rejected at the 1% significance level. Regarding the OVX–VIX relationship, the results suggest bi-directional causality between them. Second, contrary to our

**Table 8. Lags under the different criteria for VAR models (4)–(6).** * indicates the lag order selected by the criterion.

| Lag | AIC | HQ | SC | FPE |
|---|---|---|---|---|
| 1 | -17.2325 | -17.2190 | -17.19552 | 3.28e-8 |
| 2 | -17.4113 | -17.3897 | -17.35214 | 2.74e-8 |
| 3 | -17.4331 | -17.4035 | -17.35181 | 2.68e-8 |
| 4 | -17.4366 | -17.3989 | -17.3331 | 2.67e-8 |
| 5 | -17.4447 | -17.3989 | -17.31901 | 2.65e-8 |
| 6 | -17.4454 | -17.3915 | -17.29745 | 2.65e-8 |
| 7 | -17.4468* | -17.3848* | -17.27673* | 2.64e-8* |
| 8 | -17.4444 | -17.3744 | -17.25215 | 2.65e-8 |
| 9 | -17.4463 | -17.3681 | -17.23182 | 2.64e-8 |
| 10 | -17.4413 | -17.3551 | -17.2046 | 2.66e-8 |

**Table 9. TY granger causality test.**

| Relation | Chi-square | Probability |
|---|---|---|
| $\Delta\log$ (OVX) $\rightarrow$ $\Delta\log$ (VIX) | 21.1488[‡] | 0.0035 |
| $\Delta\log$ (VIX) $\rightarrow$ $\Delta\log$ (OVX) | 20.5662[‡] | 0.0044 |
| $\Delta\log$ (OVX) $\rightarrow$ $\Delta\log$ (VKOSPI) | 3.5194 | 0.8331 |
| $\Delta\log$ (VKOSPI) $\rightarrow$ $\Delta\log$ (OVX) | 16.2632[†] | 0.0228 |
| $\Delta\log$ (VIX) $\rightarrow$ $\Delta\log$ (VKOSPI) | 474.8223[‡] | $\leq$ 0.0000 |
| $\Delta\log$ (VKOSPI) $\rightarrow$ $\Delta\log$ (VIX) | 18.3266[†] | 0.0105 |

expectations, the OVX had little to do with the changes in the VKOSPI. Rather, "no Granger causality from total VKOSPI to OVX" can be rejected at the 5% significance level. Because the results differ from our expectations, we examine the relationship between these two indices over time in the following sections. Third, the null hypothesis of "no Granger causality from VIX to VKOSPI" and "no Granger causality from VKOSPI to VIX" can be rejected at the 1% and 5% significance levels, respectively. If we draw our conclusion using the 1% significance level, there is unidirectional causality from the VIX to the VKOSPI.

## BEKK-GARCH(1,1) model

We estimate the bivariate BEKK-GARCH (1,1) model's parameters by using the maximum likelihood method and the estimation results are given in Table 10.

In this study, we concentrate on the estimation results of the off-diagonal elements in matrices $A$ and $B$. The significant off-diagonal elements($a_{12}, a_{21}, a_{13}, a_{31}, a_{23}, and a_{32}$) indicate cross-volatility shock spillovers effects; whereas, the off-diagonal elements ($b_{12}, b_{21}, b_{13}, b_{31}, b_{23}, and b_{32}$) imply cross-volatility volatility spillover effects.

In terms of shock spillover effects, the off-diagonal elements $a_{21}, a_{23}, a_{31}$ and $a_{32}$ are significantly different from zero. In the case of volatility spillovers, the off-diagonal elements $b_{12}, b_{21}, b_{23}, b_{31}$ and $b_{32}$ are significantly different from zero.

These results indicate several things. First, the VIX and VKOSPI have a bi-directional spillover effect according to the elements $a_{23}, a_{32}, b_{23}$ and $b_{32}$. These results are consistent with the causality test results in Table 9.

Second, the elements $a_{21}, b_{12}$ and $b_{21}$ suggest that OVX and VIX have an influence on each other. However, the element $a_{12}$ is not significantly estimated in the BEKK model.

Third, the elements $a_{31}$ and $b_{31}$ are significantly estimated while $a_{13}$ and $b_{13}$ are not significant. This means that OVX does not affect the VKOSPI's change. These results are also consistent with the causality test results in Table 9.

## Sub-period analysis

In addition to investigating the relationship between the volatility indices for the entire period (2009–2018), the total sample is examined for structural breaks in OVX by using the [110] sequential breakpoint tests. According to the breakpoint tests, the entire sample is split into two sub-periods after locating the date of 10/8/2014 as the breakpoint. Therefore, we analyze two sub-periods, namely January 2, 2009–October 7, 2014 (sub-period 1), and October 8, 2014–December 28, 2018 (sub-period 2). The OVX time series for the two sub-periods are illustrated in Fig 2 with the breakpoint. This sub-period analysis is often carried out in other studies ([9, 27, 74, 111]).

In each sub-period, we repeatedly implement the ARDL bounds and TY Granger causality tests to examine the cointegration and direction of causality between the volatility indices.

**Table 10. BEKK model parameter estimates for the volatility indices.** The standard errors of the estimated parameters are displayed in parentheses.

| C (3 × 3) | | | A (3 × 3) | | | B (3 × 3) | | |
|---|---|---|---|---|---|---|---|---|
| $0.0273^{\ddagger}$ (0.0040) | $0.0165^{\ddagger}$ (0.0084) | $-0.0062$ (0.0046) | $0.3082^{\ddagger}$ (0.0298) | $0.0113$ (0.0345) | $0.0075$ (0.0214) | $0.7890^{\ddagger}$ (0.0676) | $0.4200^{\ddagger}$ (0.1320) | $-0.0508$ (0.0898) |
| $0$ | $0.0390^{\ddagger}$ (0.0052) | $0.0125$ (0.0068) | $-0.0636^{\ddagger}$ (0.0172) | $-0.1348^{\ddagger}$ (0.0358) | $0.3354^{\ddagger}$ (0.0224) | $-0.0029$ (0.0211) | $0.3868^{\ddagger}$ (0.0733) | $0.5360^{\ddagger}$ (0.0557) |
| $0$ | $0$ | $0.0070$ (0.0101) | $0.1082^{\ddagger}$ (0.0246) | $0.1701^{\ddagger}$ (0.0634) | $0.1274^{\ddagger}$ (0.0414) | $-0.1448^{\ddagger}$ (0.0605) | $-1.1093^{\ddagger}$ (0.0826) | $0.1573^{\ddagger}$ (0.0193) |

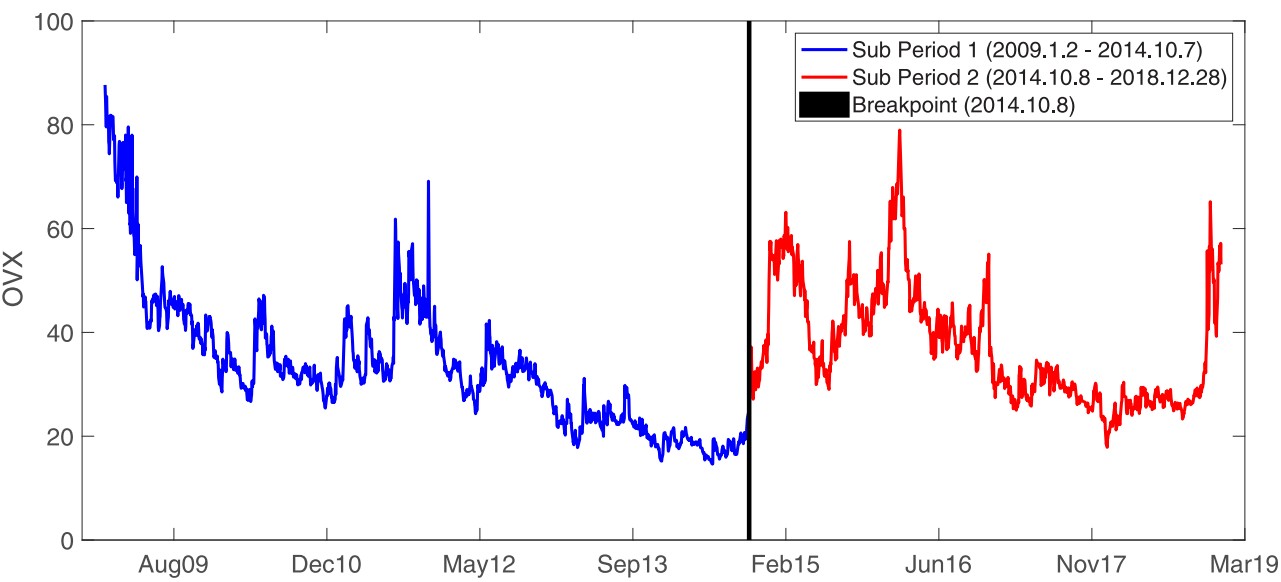

**Fig 2. The sub-periods and breakpoint for the OVX time series.**

This sub-period analysis aims to identify changes in the relationship between the volatility indices over time.

Table 11 shows the results of the ARDL bounds test by sub-period. According to the calculated $F$-statistics, $F_{VIX}(VIX \mid OVX, VKOSPI)$ and $F_{VKOSPI}(VKOSPI \mid OVX, VIX)$ have increased. On the contrary, the value of $F_{OVX}(OVX \mid VIX, VKOSPI)$ has decreased. Hence, although there are long-term relationships among the volatility indices in both sub-periods, the size of the cointegration changes over time.

Table 12 provides the results of the VAR model for the sub-period samples. Although there is bi-directional causality between the OVX and VIX in sub-period 1, it is hard to state that causation exists in sub-period 2. In the OVX–VKOSPI relationship, the null hypothesis of "no Granger causality from OVX to VKOSPI" is rejected at the 5% significance level in sub-period 1. It seems that the OVX Granger-causes VKOSPI, but not the other way around in sub-period

**Table 11. Results of the ARDL bounds tests by sub-period.**

| | Sub-period 1 | | |
|---|---|---|---|
| Model | Optimal lag ($p, q, r$) | $F$-statistic | Decision |
| $F_{OVX}(OVX \mid VIX, VKOSPI)$ | (1,1,1) | 10.7566 | Long-term relationship exists |
| $F_{VIX}(VIX \mid OVX, VKOSPI)$ | (1,1,1) | 13.6964 | Long-term relationship exists |
| $F_{VKOSPI}(VKOSPI \mid OVX, VIX)$ | (1,3,1) | 12.7507 | Long-term relationship exists |
| | Sub-period 2 | | |
| Model | Optimal lag ($p, q, r$) | $F$-statistic | Decision |
| $F_{OVX}(OVX \mid VIX, VKOSPI)$ | (1,1,3) | 6.8848 | Long-term relationship exists |
| $F_{VIX}(VIX \mid OVX, VKOSPI)$ | (1,1,1) | 15.48 | Long-term relationship exists |
| $F_{VKOSPI}(VKOSPI \mid OVX, VIX)$ | (1,2,1) | 19.7431 | Long-term relationship exists |
| Critical values | 1% | 5% | 10% |
| Lower bounds $I(0)$ | 5.15 | 3.79 | 3.17 |
| Upper bounds $I(1)$ | 6.36 | 4.85 | 4.14 |

**Table 12. Results of the TY granger causality tests by sub-period.** The optimal lag is $m = 2$ for both sub-periods.

| Relation | Sub-period 1 | | Sub-period 2 | |
|---|---|---|---|---|
| | Chi-square | Probability | Chi-square | Probability |
| $\Delta\log(OVX) \rightarrow \Delta\log(VIX)$ | $10.3075^{\ddagger}$ | 0.0058 | 2.4829 | 0.2890 |
| $\Delta\log(VIX) \rightarrow \Delta\log(OVX)$ | $21.8181^{\ddagger}$ | $\leq 0.0000$ | 5.7428 | 0.0566 |
| $\Delta\log(OVX) \rightarrow \Delta\log(VKOSPI)$ | $6.6280^{\dagger}$ | 0.0364 | 1.2069 | 0.5469 |
| $\Delta\log(VKOSPI) \rightarrow \Delta\log(OVX)$ | 2.9162 | 0.2327 | 5.3936 | 0.0674 |
| $\Delta\log(VIX) \rightarrow \Delta\log(VKOSPI)$ | $208.7702^{\ddagger}$ | $\leq 0.0000$ | $261.4142^{\ddagger}$ | $\leq 0.0000$ |
| $\Delta\log(VKOSPI) \rightarrow \Delta\log(VIX)$ | 5.9136 | 0.0520 | 1.8910 | 0.3885 |

**Table 13. BEKK model parameter estimates for the volatility indices by sub-period.**

Sub-period 1

C (3 × 3):
$$\begin{pmatrix} -0.0425^{\ddagger}_{(0.0008)} & -0.0379^{\ddagger}_{(0.0021)} & -0.0033_{(0.0024)} \\ 0 & -0.0523^{\ddagger}_{(0.0017)} & -0.0170^{\ddagger}_{(0.0018)} \\ 0 & 0 & -0.038^{\ddagger}1_{(0.0013)} \end{pmatrix}$$

A (3 × 3):
$$\begin{pmatrix} 0.3593^{\ddagger}_{(0.0349)} & 0.1364^{\ddagger}_{(0.0377)} & 0.0828^{\ddagger}_{(0.0258)} \\ -0.2154^{\ddagger}_{(0.0354)} & -0.5036^{\ddagger}_{(0.0571)} & 0.2850^{\ddagger}_{(0.0393)} \\ 0.1184^{\ddagger}_{(0.0330)} & 0.2253^{\ddagger}_{(0.0511)} & 0.1071^{\ddagger}_{(0.0259)} \end{pmatrix}$$

B (3 × 3):
$$\begin{pmatrix} -0.2062_{(0.1906)} & 0.0000_{(0.0584)} & -0.1315^{\ddagger}_{(0.0357)} \\ 0.0804_{(0.1567)} & 0.0012_{(0.0620)} & 0.0884_{(0.1307)} \\ -0.3118^{\dagger}_{(0.1411)} & -0.0001_{(0.0721)} & -0.2161_{(0.1363)} \end{pmatrix}$$

Sub-period 2

C (3 × 3):
$$\begin{pmatrix} 0.0471^{\ddagger}_{(0.0013)} & 0.0321^{\ddagger}_{(0.0024)} & 0.0102^{\ddagger}_{(0.0017)} \\ 0 & 0.0784^{\ddagger}_{(0.0020)} & 0.0139^{\ddagger}_{(0.0019)} \\ 0 & 0 & 0.0461^{\ddagger}_{(0.0016)} \end{pmatrix}$$

A (3 × 3):
$$\begin{pmatrix} 0.3165^{\ddagger}_{(0.0641)} & 0.0140_{(0.0911)} & -0.0799_{(0.0594)} \\ -0.0056_{(0.0219)} & -0.0307_{(0.0477)} & 0.4336^{\ddagger}_{(0.0353)} \\ -0.0111_{(0.0330)} & 0.2813^{\ddagger}_{(0.0681)} & -0.1972^{\ddagger}_{(0.0519)} \end{pmatrix}$$

B (3 × 3):
$$\begin{pmatrix} -0.0070_{(0.0521)} & -0.0277_{(0.0994)} & -0.0001_{(0.1021)} \\ -0.0071_{(0.1226)} & -0.0278_{(0.1363)} & -0.0040_{(0.2680)} \\ -0.0098_{(0.0423)} & -0.0442_{(0.0925)} & -0.0051_{(0.1483)} \end{pmatrix}$$

1 but Granger-causality is not found in sub-period 2. In the VIX–VKOSPI relationship, the TY Granger causality test results indicate that the VIX has an enormous influence on the VKOSPI over time.

We strengthen the results of the TY Granger causality tests by using the BEKK-GARCH model and the estimation results for the BEKK-GARCH model are given in Table 13. From the results, the following facts can be found: (1) We find a strong bi-directional shock spillover between the OVX and VIX in sub-period 1 but the bi-directional shock spillover does not occur in sub-period 2. (2) This relationship change happens similarly between the OVX and VKOSPI. (3) The VIX and VKOSPI have a strong bi-directional relation regardless of the periods. Furthermore, the value of $a_{23}$ rises from 0.2850 to 0.4336 through sub-period 1 and 2, which means that the transmissions from the VIX to VKOSPI in sub-period 2 are stronger than those in sub-period 1.

## Discussion in terms of shale gas and risk management

As mentioned above, several events have caused a large price change in the crude oil market. Among these, the advent of the shale gas revolution has directly affected oil prices and revolutionized the U.S. energy sector, including prices and consumption ([112]). Many studies on the effect of shale gas have been reported ([113], [114], [115], [116], [117]).

(a)

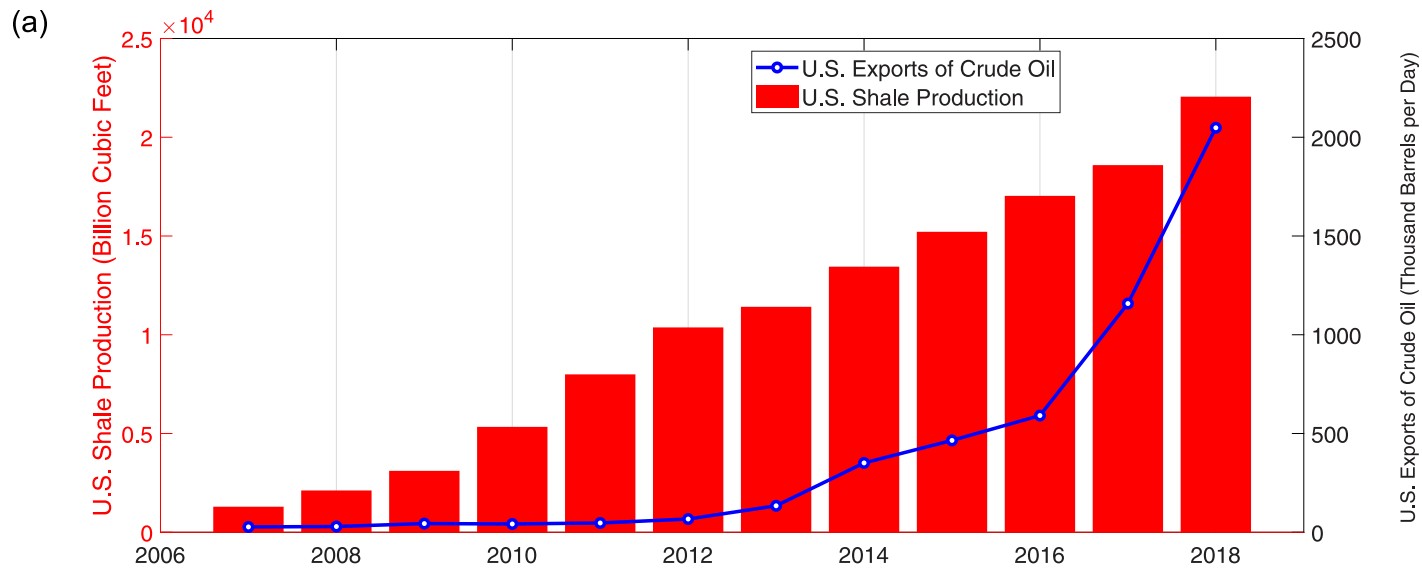

(b)

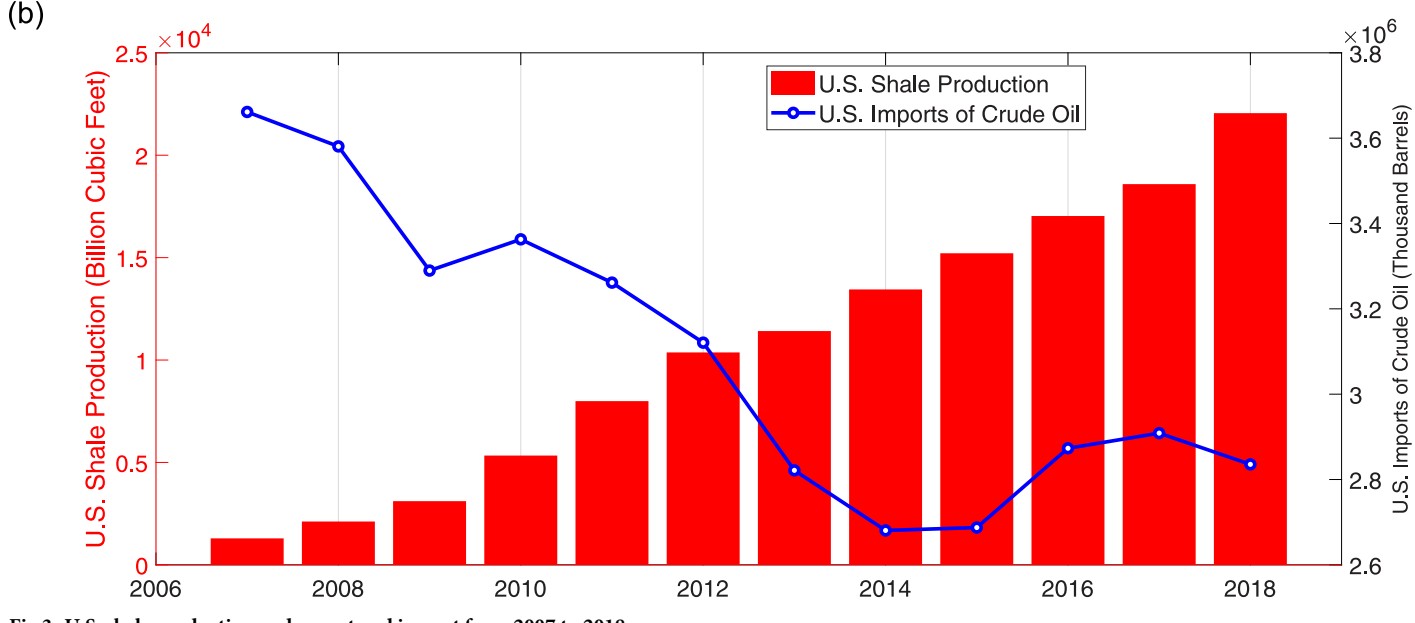

**Fig 3. U.S. shale production and export and import from 2007 to 2018.**

The Fig 3 illustrates the annual shale gas production, the exports and imports of crude oil data(Source: U.S. Energy Information Administration). From the figure, as shale gas production grows, exports of crude oil definitely increase. By contrast, imports of crude oil decrease dramatically. This fact suggests that shale gas production has affected the import and export of crude oil.

In the previous section, we use the two sub-periods, January 2, 2009–October 7, 2014 (sub-period 1), and October 8, 2014–December 28, 2018 (sub-period 2). According to [116], the price of oil experienced one of its largest declines in modern history between June 2014 and December 2014. Therefore, we can regard these two sub-periods as the period during which

the shale gas revolution took place (sub-period 2) and the period before it happened (sub-period 1), respectively.

The following conclusions are obtained by interpreting the sub-period empirical results shown in Section 5 in terms of the shale gas.

In sub-period 1, that is, in times of low shale gas production, the U.S. stock and crude oil markets are closely linked according to the sub-period analysis, whose results are consistent with several studies ([118], [119], [120], [121], [122]). Meanwhile, because of the increase in production of shale gas in sub-period 2, dependence on crude oil imports has decreased, and the bi-directional relationship between the stock market and the oil market has disappeared.

Of course, the shale gas revolution is not the only direct cause of the change in the relationship between the OVX and VIX. There are many factors that have affected the relationship between the two markets, but what we want to argue here is that shale gas is the one of main factors that has caused the change. As you can see from Fig 3, the shale gas revolution would have had a major effect on the crude oil market and on the oil-related market and we regard the effect as a change in the relationship between the OVX and VIX. The arguments made through this research process can be found in other papers. To study the changes according to the shale gas production, there are several studies that have implemented sub-period analysis. [123] examines the effect of the shale gas revolution on North American and European natural gas markets through two sub-periods, the pre-revolution period and the post-revolution period. Similarly, [124] investigate the effects of oil price shocks on the stock returns in the oil industrial chain companies and inspect the differences between the two periods. Although [124] focus on the oil price shock itself, they explain the shale gas solution as the cause of the oil price shock.

In the Introduction, we said that checking the volatility of crude oil prices and establishing an appropriate strategy is important for risk management. In terms of the management of derivative asset portfolios, volatility risk plays a crucial role. Therefore, in order to manage the portfolios of crude oil or its derivatives, we need to examine the volatility of crude oil, or the OVX, carefully. Research on portfolio risk management using volatility indices continues to be reported ([46], [47], [48]). However, it mainly uses the VIX to investigate hedging ability.

Risk management for crude oil has also been studied and these studies focus on price risk not volatility risk. For example, there are studies that calculate the optimal portfolio weight or the hedge ratio of a crude oil portfolio using several models, such as GARCH and BEKK models ([125], [126, 127]). In addition, some studies examine the volatility spillovers between crude oil and other assets and also determine the optimal weights and hedge ratio of the portfolios according to the estimated spillover effects ([105, 121], [128]).

The difference between the existing studies and this study is that we use volatility indices to identify the relationship between crude oil and the stock market. The volatility index represents a measure of the risk and it implies the market participants' expectations for the market. Therefore, we can see whether the expected risk predicted by each market participant affects the expectations of other market participants by looking at the changes in the relationship between the volatility indices. Our empirical results indicate that forecasts of risks to crude oil and risks to the stock market were mutually influenced prior to 2014, but recently the effects have been reduced. Furthermore, we explain the relationships between the U.S. and South Korea stock markets, and between crude oil and the South Korean stock market with respect to the volatility indices. The studies that examine the relationship between crude oil and U.S. stock markets are quoted in Section 2. There are also studies on the effect of changes in crude oil prices on the South Korean economy ([129], [130], [131]), but they use crude oil prices and the South Korean stock market index not volatility indices. Likewise, [132] and [133] examine

the relationship between stock prices in the U.S. and South Korean stock markets, but they use stock indices.

We can manage the volatility risk arising from the crude oil price fluctuations by calculating optimal portfolio weights and hedge ratios. Assume that an investor attempts to hedge exposure to crude oil price fluctuation for a portfolio of oil and stocks. Then, the investor wants to minimize the risk of his/her oil-stock portfolio without reducing its expected returns. According to [33], conditional volatilities can be used to construct optimal portfolio weights

$$
w_t^{S0} = \frac{h_t^{S} - h_t^{S0}}{h_t^{O} - 2h_t^{SO} + h_t^{S}}, \quad \text{with} \quad w_t^{S0} = \begin{cases} 0, & \text{if } w_t^{S0} < 0 \\ w_t^{S0}, & \text{if } 0 \geq w_t^{S0} \leq 1 \\ 1, & \text{if } w_t^{S0} > 1 \end{cases}
$$

where $w_t^{S0}$ is the weight of oil in the crude oil and stock portfolio at time $t$ and $h_t^{S}, h_t^{O}$, and $h_t^{SO}$ are the conditional volatility of oil and stock and conditional covariance between oil and stock returns at time $t$, respectively. Therefore, when we calculate the conditional volatilities by using some models, such as the GARCH and BEKK models, we obtain a dynamic hedge ratio. Furthermore, given an OVX derivatives and stocks portfolio, we can also apply this process to calculate the optimal hedge ratio.

## Concluding remarks

This study investigates the relationship between the uncertainties in oil prices and stock markets (United States and South Korea). The motivation for conducting this study is the lack of empirical results on the relationship between the volatility of oil prices and of the U.S. and South Korean stock markets. South Korea relies on imports for most of its crude oil consumption, which implies that the country's economic status might be affected by a change in oil prices. Another motivation is South Korea's dependence on the United States. South Korea is one of the United States' most important strategic and economic partners in Asia. Therefore, we aimed to confirm the extent to which uncertainty in the U.S. stock market affects uncertainty in the South Korean stock market.

We address this issue by applying two frequently used methods, namely the ARDL bounds test and TY Granger causality test, to three volatility indices (the OVX, VIX, and VKOSPI). Additionally, we investigate whether the relationship between them changes over time through a sub-period analysis. In order to enhance the robustness of the test results, we employ the BEKK-GARCH model. The empirical results provide a number of interesting conclusions with useful practical implications. Our main findings can be summarized as follows.

First, the results of the ARDL bounds tests indicate that there is a long-run relationship between the oil and stock market implied volatility indices. The sub-period analysis results also suggest that this relationship is constant.

Second, we find bi-directional Granger causality between the OVX and VIX for the whole sample period, which is consistent with [67]'s results. However, the sub-period analysis finds no statistically significant Granger causality between them in sub-period 2. This finding suggests that the influence between the oil and U.S. stock markets has changed dramatically because of a certain event. Many studies show that this event is related to the shale gas revolution ([113], [134], [135], [136], [137], [138], [112], [139]).

Third, through the sub-period analysis, we confirm that the relationship between the OVX and VKOSPI has changed over time. The OVX Granger causes the VKOSPI during sub-period 1, although there is no Granger causality between them during sub-period 2. These changes

may have occurred because of a number of factors; however, one of the main factors is the effect of the shale gas revolution, as noted above.

Lastly, according to the results of the TY Granger causality test in the sub-period analysis, there is unidirectional Granger causality from the VIX to VKOSPI in both sub-periods. In other words, shocks to the U.S. market also affect the South Korean market significantly and this phenomenon has continued for the past decade.

To summarize, the future expectations of the U.S. and South Korean stock markets are less sensitive to the risk of crude oil fluctuations than before, and shale gas production in the United States may be one of reasons for this decrease in sensitivity. Furthermore, the influence of the U.S. stock market on the South Korean stock market has increased. That is, future expectations of the U.S. market have a significant effect on predictions for the South Korean market.

This study is noteworthy in that the influence of crude oil volatility on the U.S. and South Korean stock markets has decreased significantly. In addition, it is remarkable that the influence of the U.S. stock market on the South Korean stock market has increased.

We present some applications and implications based on this study's results. First, for both investors and policymakers, the key application of our work is properly forecasting financial market volatility. In particular, changes in the VIX in the U.S. stock market are strongly related to those in the South Korean stock market. In other words, we can increase the predictive power of the future VKOSPI in the South Korean stock market using the movement of the VIX. As [140] claim, forecasting volatility indices may be more beneficial to the decision-making of all stock market participants (including financial traders, manufacturers, and policymakers). Second, because volatility indices are used to hedge volatility risk, our findings will help to manage volatility risk in crude oil portfolios. According to [141] and [67], OVX derivatives, futures, and options can be a financial tool to hedge volatility risk. Furthermore, volatility indices have become a popular asset class for investors considering diversifying their portfolio strategy. Thus, our empirical findings can be used to examine and evaluate volatility derivatives, such as OVX options and futures. Third, contrary to expectations, South Korea, an emerging market, has not been sensitive to crude oil risks lately. There are many reasons for this, but oil import diversification may be one of them. Therefore, this study can be seen as evidence of the effect of crude oil import diversification for oil-importers, and in particular, South Korea.

Possible future studies include research on optimal portfolio weights and hedge ratios with respect to the sub-period data used in this study. In addition, we can consider the causality between the positive and negative shocks of volatility indices by using nonlinear causality tests. In other words, it is necessary to ascertain how they affect and receive each other when market risk increases and when it decreases.

## Acknowledgments

The authors thank the editor Stefan Cristian Gherghina, and two referees for useful suggestions. The authors thank Prof. Bong-Gyu Jang for useful comments.

## Author Contributions

**Formal analysis:** Sun-Yong Choi.

**Funding acquisition:** Sun-Yong Choi.

**Investigation:** Changsoo Hong.

**Methodology:** Sun-Yong Choi.

**Resources:** Changsoo Hong.

**Software:** Sun-Yong Choi.

**Validation:** Sun-Yong Choi, Changsoo Hong.

**Visualization:** Sun-Yong Choi.

**Writing – original draft:** Sun-Yong Choi.

**Writing – review & editing:** Sun-Yong Choi, Changsoo Hong.

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
