## [Decision Letter · Decision Letter 0]

13 Jan 2020

PONE-D-19-34694

Relationship between uncertainty in the oil and stock markets before and after the shale gas revolution: Evidence from the OVX, VIX, and VKOSPI volatility indices

PLOS ONE

Dear Prof. choi,

Thank you for submitting your manuscript to PLOS ONE. After careful consideration, we feel that it has merit but does not fully meet PLOS ONE’s publication criteria as it currently stands. Therefore, we invite you to submit a revised version of the manuscript that addresses the points raised during the review process.

Further revisions are required regarding, even not limited to, research motivation and study contribution, discussion concerning prior studies, outcomes’ implications, quantitative methodology argumentation.

We would appreciate receiving your revised manuscript by Feb 27 2020 11:59PM. To enhance the reproducibility of your results, we recommend that if applicable you deposit your laboratory protocols in protocols.io, where a protocol can be assigned its own identifier (DOI) such that it can be cited independently in the future. For instructions see: http://journals.plos.org/plosone/s/submission-guidelines#loc-laboratory-protocols

We look forward to receiving your revised manuscript.

Kind regards,

Stefan Cristian Gherghina, PhD

Academic Editor

PLOS ONE

Journal Requirements:

"NO authors have competing interests"

We note that one or more of the authors are employed by a commercial company: "NICE Pricing & Information Inc."

Reviewers' comments:

Reviewer's Responses to Questions

**Comments to the Author**

1. Is the manuscript technically sound, and do the data support the conclusions?

Reviewer #1: Yes

Reviewer #2: Yes

2. Has the statistical analysis been performed appropriately and rigorously? 

Reviewer #1: No

Reviewer #2: Yes

3. Have the authors made all data underlying the findings in their manuscript fully available?

Reviewer #1: Yes

Reviewer #2: No

4. Is the manuscript presented in an intelligible fashion and written in standard English?

Reviewer #1: Yes

Reviewer #2: Yes

5. Review Comments to the Author

Reviewer #1: Comments

1. Aside from revealing the lead-lag relationship among the variables employed in this study, authors had better mention why this study is important and why this study are able to contribute to the existing studies in the abstract section of this paper.

2. Authors mention that “observing oil price volatility and taking actions in response to its expected changes are essential for managing risk” would be important for the relevant literature. We argue that the above concern might be already taken into account in the relevant studies. As a result, we document that authors have to mention the main differences between your study and other relevant studies in detail. In addition, how to manage the risks in terms of oil price fluctuated considerably should be illustrated in detail as well.

3. The motivation of this study should be enhanced and appeared in the front of this paper since readers prefer the strong motivation presented in the beginning of the introduction section.

4. Authors had better provide the reasons why authors adopt these methodologies (i.e. the autoregressive distributed lag (ARDL) bounds test of cointegration as well as the Toda–Yamamoto (TY) Granger causality test of Toda and Yamamoto (1995)). Why not employing other methodologies due to one of the methodologies proposed in 1995? Are there any other update and appropriate methodologies likely employed for this paper?

5. In fact, ARCH effects are likely existed in financial time series. We suggest that authors have better employ Multivariate GARCH-family models to examine the relationships among the variables (i.e. OVX, VIX, and VKOSPI) employed in this study for robustness.

6. The contribution of this study could be enhanced since the presentation of the contribution of this study might not appeal readers.

7. Due to the data period employing in this study from January 2, 2009 to December 28, 2018, we argue that authors had better to examine whether there are any structural changes existed over the data period instead of examining the relationship among these variables by using two almost equal sub-periods (i.e. two sub-periods, namely January 2009–May 2014 (sub-period 1) and June 2014–December 2018 (sub-period 2).

Reviewer #2: Journal Title: PLOS ONE

Manuscript #: PONE-D-19-34694

Paper Title: Relationship between uncertainty in the oil and stock markets before and after the shale gas revolution: Evidence from the OVX, VIX, and VKOSPI volatility indices

This paper studies the long-term relationships among the volatility indices of OVX, VIX, and VKOSPI while shale gas revolution happens and use the methodologies of ARDL and Toda-Yamamoto granger causality test. The results show that there is a bidirectional causality between OVX and VIX with the shale gas revolution and also finds some unidirectional causalities from VIX to VKOSPI. The topic is interesting but the current quality of the paper below the minimum acceptance level required/requested by the journal. I would suggest authors to improve the paper in accordance with the comments below.

Comments

1. First of all, this paper lacks solid theoretical backgrounds and motivations. Most of the journal readers with finance/economics backgrounds might understand that the oil and stock markets would affect with each other but, for other readers in different areas might need to learn more such relationships. A brief discussion as I mentioned above should be added in the paper. After that, authors should further develop the relationships among these volatility indices.

2. Authors should add a section of Literature Review to comprehensively review the relationships among these volatility indices.

3. In the current version, I did not see the discussion of shale gas revolutions and have no idea how authors define this revolution. In addition, authors should provide more explanations why the sample period is divided into two sub-periods.

4. Several research might be included in the paper as follows:

Giot, P. (2005). Relationships between implied volatility indexes and stock index returns. The Journal of Portfolio Management, 31(3), 92-100.

Chiang, S. M., Chen, C. D., & Huang, C. M. (2019). Analyzing the impacts of foreign exchange and oil price on biofuel commodity futures. Journal of International Money and Finance, 96, 37-48.

5. I would also like to see more financial/economic applications and implications based on the results found in the paper.

6. PLOS authors have the option to publish the peer review history of their article (what does this mean?). If published, this will include your full peer review and any attached files.

Reviewer #1: No

Reviewer #2: No

---

## [Author Response · Author response to Decision Letter 0]

25 Feb 2020

We agree with the reviewer's suggestion. We have revised the manuscript accordingly.

---

## [Decision Letter · Decision Letter 1]

9 Mar 2020

PONE-D-19-34694R1

Relationship between uncertainty in the oil and stock markets before and after the shale gas revolution: Evidence from the OVX, VIX, and VKOSPI volatility indices

PLOS ONE

Dear Prof. choi,

Thank you for submitting your manuscript to PLOS ONE. After careful consideration, we feel that it has merit but does not fully meet PLOS ONE’s publication criteria as it currently stands. Therefore, we invite you to submit a revised version of the manuscript that addresses the points raised during the review process.

Further revisions are required concerning study motivation, research contribution, as well as empirical methodology and findings.

We would appreciate receiving your revised manuscript by Apr 23 2020 11:59PM. To enhance the reproducibility of your results, we recommend that if applicable you deposit your laboratory protocols in protocols.io, where a protocol can be assigned its own identifier (DOI) such that it can be cited independently in the future. For instructions see: http://journals.plos.org/plosone/s/submission-guidelines#loc-laboratory-protocols

We look forward to receiving your revised manuscript.

Kind regards,

Stefan Cristian Gherghina, PhD. Habil.

Academic Editor

PLOS ONE

Reviewers' comments:

Reviewer's Responses to Questions

**Comments to the Author**

1. If the authors have adequately addressed your comments raised in a previous round of review and you feel that this manuscript is now acceptable for publication, you may indicate that here to bypass the “Comments to the Author” section, enter your conflict of interest statement in the “Confidential to Editor” section, and submit your "Accept" recommendation.

Reviewer #1: All comments have been addressed

Reviewer #2: All comments have been addressed

2. Is the manuscript technically sound, and do the data support the conclusions?

Reviewer #1: No

Reviewer #2: Yes

3. Has the statistical analysis been performed appropriately and rigorously? 

Reviewer #1: Yes

Reviewer #2: Yes

4. Have the authors made all data underlying the findings in their manuscript fully available?

Reviewer #1: Yes

Reviewer #2: Yes

5. Is the manuscript presented in an intelligible fashion and written in standard English?

Reviewer #1: Yes

Reviewer #2: Yes

6. Review Comments to the Author

Reviewer #1: Comments

Please point-by-point response to the comments proposed in the previous review in detail as shown below; otherwise, this paper might not be accepted.

1. Aside from revealing the lead-lag relationship among the variables employed in this study, authors had better mention why this study is important and why this study are able to contribute to the existing studies in the abstract section of this paper.

2. Authors mention that “observing oil price volatility and taking actions in response to its expected changes are essential for managing risk” would be important for the relevant literature. We argue that the above concern might be already taken into account in the relevant studies. As a result, we document that authors have to mention the main differences between your study and other relevant studies in detail. In addition, how to manage the risks in terms of oil price fluctuated considerably should be illustrated in detail as well.

3. The motivation of this study should be enhanced and appeared in the front of this paper since readers prefer the strong motivation presented in the beginning of the introduction section.

4. Authors had better provide the reasons why authors adopt these methodologies (i.e. the autoregressive distributed lag (ARDL) bounds test of cointegration as well as the Toda–Yamamoto (TY) Granger causality test of Toda and Yamamoto (1995)). Why not employing other methodologies due to one of the methodologies proposed in 1995? Are there any other update and appropriate methodologies likely employed for this paper?

5. In fact, ARCH effects are likely existed in financial time series. We suggest that authors have better employ Multivariate GARCH-family models to examine the relationships among the variables (i.e. OVX, VIX, and VKOSPI) employed in this study for robustness.

6. The contribution of this study could be enhanced since the presentation of the contribution of this study might not appeal readers.

7. Due to the data period employing in this study from January 2, 2009 to December 28, 2018, we argue that authors had better to examine whether there are any structural changes existed over the data period instead of examining the relationship among these variables by using two almost equal sub-periods (i.e. two sub-periods, namely January 2009–May 2014 (sub-period 1) and June 2014–December 2018 (sub-period 2).

Decision: Major revision

Reviewer #2: No further comments and I believe authors have improved the paper in accordance with the comments. I would recommend the paper to be accepted and published by the journal.

7. PLOS authors have the option to publish the peer review history of their article (what does this mean?). If published, this will include your full peer review and any attached files.

Reviewer #1: No

Reviewer #2: Yes: Chunda Chen

---

## [Author Response · Author response to Decision Letter 1]

11 Mar 2020

We address the Reviewers’ comments and list of changes that we made to our manuscript according to their reports. The original Reviewers’ comments are provided in bold fonts. Our answers are described in a point-to-point manner starting with a bullet. Please see the attached file “Response to Reviewers” for your review. 

#Comment 1. Aside from revealing the lead-lag relationship among the variables employed in this study, authors had better mention why this study is important and why this study are able to contribute to the existing studies in the abstract section of this paper.

-> Thank you for your valuable comment. As requested, we mentioned the importance and contribution to the existing studies in the abstract. The following sentence are added:

Abstract section, Page 1 : "In light of crude oil’s essential role in the world economy, the relationship between crude oil prices and stock markets is of great interest to finance practitioners and researchers. Previous studies usually use crude oil and stock price data to examine this relationship. This study contributes to the literature by investigating the links between crude oil and stock markets in terms of volatility indices. ... These results have important implications for the analysis of portfolio risk management and for assisting energy policymakers and traders in making effective decisions and investments, respectively" 

#Comment 2. Authors mention that “observing oil price volatility and taking actions in response to its expected changes are essential for managing risk” would be important for the relevant literature. We argue that the above concern might be already taken into account in the relevant studies. As a result, we document that authors have to mention the main differences between your study and other relevant studies in detail. In addition, how to manage the risks in terms of oil price fluctuated considerably should be illustrated in detail as well.

-> Thank you for your valuable comment. As requested, we mentioned the main differences between our study and other relevant studies in detail. The following sentence are added to the Introduction, Section Discussion in terms of shale gas and risk management and Section Concluding remarks:

Introduction section, Page 3, lines 79 - 86 : 

Discussion in terms of shale gas and risk management section, Page 19 - 20, lines 472 - 485 : 

Concluding remarks section, Page 21, lines 536 - 539 : 

Furthermore, we added a section(Section Discussion in terms of shale gas and risk management) to state how to manage the risks in terms of oil price fluctuated in detail. In the section, we illustrated a way to manage the risk of oil price fluctuation for a portfolio of oil and stocks, according to the relevant literature. We mentioned the calculation of optimal portfolio weights and hedge ratios as a future study. The following sentence are added: 

Discussion in terms of shale gas and risk management section, Page 20, lines 487 – 493: 

Concluding remarks section, Page 21, lines 558 – 559: 

#Comment 3. The motivation of this study should be enhanced and appeared in the front of this paper since readers prefer the strong motivation presented in the beginning of the introduction section. 

-> Thank you for your valuable comment. As requested, we enhanced the motivation of this study and presented it in the Introduction section. The following sentence are added: 

Introduction section, Page 2, lines 42 – 51: 

#Comment 4. Authors had better provide the reasons why authors adopt these methodologies (i.e. the autoregressive distributed lag (ARDL) bounds test of cointegration as well as the TodaYamamoto (TY) Granger causality test of Toda and Yamamoto (1995)). Why not employing other methodologies due to one of the methodologies proposed in 1995? Are there any other update and appropriate methodologies likely employed for this paper? 

-> Thank you for your valuable comment. As requested, we introduced the cointegration test and mentioned the reason why we adopt the ARDL bounds test. About the ARDL bounds test, the following sentence are added: 

ARDL bounds tests subsection, Page 9 – 10, lines 199 – 208 :

About TY Granger causality test, we introduced the causality test and mentioned the reason why we adopt the TY Granger causality test. Furthermore, we cited some studies in which TY causality test is used. We emphasize the following sentence: 

TY Granger causality tests subsection, Page 11, lines 236 – 247: 

TY Granger causality tests subsection, Page 11, lines 264 – 274: 

In addition, we referred to developed methodologies for Causality test since TY Granger causality test of Toda and Yamamoto (1995)). They are nonlinear version, asymmetric, and bootstrap panel causality tests. We mentioned the nonlinear causality tests as a future study. The following sentence are added: 

TY Granger causality tests subsection, Page 11, lines 254 – 263: 

Concluding remarks section, Page 21, lines 559 – 562: 

#Comment 5. In fact, ARCH effects are likely existed in financial time series. We suggest that authors have better employ Multivariate GARCH-family models to examine the relationships among the variables (i.e. OVX, VIX, and VKOSPI) employed in this study for robustness. 

-> Thank you for your valuable comment. As requested, we employ BEKK-GARCH model as multivariate GARCH-family model to examine the relationships among volatility indices. We briefly introduced BEKK-GARCH model in subsection Multivariate GARCH model and added empirical results by the BEKK-GARCH model were given in subsection BEKK- GARCH(1,1) model and section Sub-period analysis. These results are also consistent with the causality test results obtained by TY Granger causality test. 

#Comment 6. The contribution of this study could be enhanced since the presentation of the contribution of this study might not appeal readers. 

-> Thank you for your valuable comment. As requested, we enhanced the contribution of this study and mentioned three main contributions in the Introduction. The following sentence are added: 

Introduction section, Page 3, lines 87 – 103: 

In order to appeal readers, we presented some applications and implications based on this empirical results. The following paragraph is added to the Section Concluding remarks: 

Concluding remarks section, Page 21, lines 540 – 557: 

#Comment 7. Due to the data period employing in this study from January 2, 2009 to December 28, 2018, we argue that authors had better to examine whether there are any structural changes existed over the data period instead of examining the relationship among these variables by using two almost equal sub-periods (i.e. two sub-periods, namely January 2009May 2014 (sub-period 1) and June 2014December 2018 (sub-period 2). 

-> As requested, we examined the structural changes over the data period and calculated structural breakpoints for OVX time series by the algorithm described in [Bai and Perron, 2003]. In addition, we referred to some studies in which the sub-period analysis was investigated. The following sentence are added: 

Sub-period analysis section, Page 16, lines 383 – 390:

---

## [Decision Letter · Decision Letter 2]

27 Mar 2020

PONE-D-19-34694R2

Relationship between uncertainty in the oil and stock markets before and after the shale gas revolution: Evidence from the OVX, VIX, and VKOSPI volatility indices

PLOS ONE

Dear Prof. choi,

Thank you for submitting your manuscript to PLOS ONE. After careful consideration, we feel that it has merit but does not fully meet PLOS ONE’s publication criteria as it currently stands. Therefore, we invite you to submit a revised version of the manuscript that addresses the points raised during the review process.

The manuscript requires further major revisions.

We would appreciate receiving your revised manuscript by May 11 2020 11:59PM. To enhance the reproducibility of your results, we recommend that if applicable you deposit your laboratory protocols in protocols.io, where a protocol can be assigned its own identifier (DOI) such that it can be cited independently in the future. For instructions see: http://journals.plos.org/plosone/s/submission-guidelines#loc-laboratory-protocols

We look forward to receiving your revised manuscript.

Kind regards,

Stefan Cristian Gherghina, PhD. Habil.

Academic Editor

PLOS ONE

Reviewers' comments:

Reviewer's Responses to Questions

**Comments to the Author**

1. If the authors have adequately addressed your comments raised in a previous round of review and you feel that this manuscript is now acceptable for publication, you may indicate that here to bypass the “Comments to the Author” section, enter your conflict of interest statement in the “Confidential to Editor” section, and submit your "Accept" recommendation.

Reviewer #1: All comments have been addressed

Reviewer #2: All comments have been addressed

2. Is the manuscript technically sound, and do the data support the conclusions?

Reviewer #1: Yes

Reviewer #2: Yes

3. Has the statistical analysis been performed appropriately and rigorously? 

Reviewer #1: No

Reviewer #2: Yes

4. Have the authors made all data underlying the findings in their manuscript fully available?

Reviewer #1: Yes

Reviewer #2: No

5. Is the manuscript presented in an intelligible fashion and written in standard English?

Reviewer #1: Yes

Reviewer #2: Yes

6. Review Comments to the Author

Reviewer #1: 1. This study contributes to the literature by investigating the links between crude oil and stock markets in terms of volatility indices.”

The above contribution might not appeal to readers. We argue that authors should mention something new and something important related to this study, which might be able to persuade readers that this study would significantly contribute to the existing literature.

2. While authors respond to these comments, readers would prefer that the adding context or revised context would be shown right below the response to comments, since readers might not prefer the presentation as shown below.

The following sentences are added:

Discussion in terms of shale gas and risk management section, Page 20, lines 487 – 493:

Concluding remarks section, Page 21, lines 558 – 559:

In addition, the same concerns for responding to other comments

Decision: Major revision

Reviewer #2: None. Authors have improved the paper in accordance with the comments. I have no further suggestions at the current stage.

7. PLOS authors have the option to publish the peer review history of their article (what does this mean?). If published, this will include your full peer review and any attached files.

Reviewer #1: No

Reviewer #2: No

---

## [Author Response · Author response to Decision Letter 2]

30 Mar 2020

We address the Reviewers’ comments and list of changes that we made to our manuscript according to their reports. The original Reviewers’ comments are provided in bold fonts. Our answers are described in a point-to-point manner starting with a bullet. Please see the attached file “Response to Reviewers” for your review.

#Comment 1. ”This study contributes to the literature by investigating the links between crude oil and stock markets in terms of volatility indices.” 

 The above contribution might not appeal to readers. We argue that authors should mention something new and something important related to this study, which might be able to persuade readers that this study would significantly contribute to the existing literature. 

-> Thank you for your valuable comment. As requested, we mentioned the importance and contribution to the existing studies in the abstract. The abstract has been modified as follows: 

Abstract: ”In light of crude oils essential role in the world economy, the relationship between crude oil prices and stock markets is of great interest to finance practitioners and researchers. Previous studies usually use crude oil and stock price data to examine this relationship. Financial markets generally view volatility index as a measure of fear in the markets and economy. Therefore, we investigate the relationship among the volatility indices of crude oil and stock in order to derive important managerial implications. Furthermore, we evaluate the change in the relationship between the volatility indices through a sub-period analysis. These are the contributions of this study. Specifically, we examine the causal relationships among the crude oil, S&P 500 index, and KOSPI 200 index volatilities, adopting the autoregressive distributed lag (ARDL) bounds test to determine the long-term relationship. Based on the cointegration results of the ARDL bounds test, we adopt the Toda–Yamamoto Granger causality test to identify the causal relationship between the variables. Furthermore, we employ a BEKK-GARCH model for enhancing the robustness of the causality test results. We also confirm the change in this relationship over time through a sub-period analysis, using periods that include and exclude the shale gas revolution. These experiments indicate that the OVX and VIX show bi-directional causality in the period that includes the shale gas revolution and no causality in the period that does not. Further, the OVX Granger causes the VKOSPI in the former period, but there is no causality between them in the latter period. Finally, we find strong uni- directional causality from the VIX to the VKOSPI in both sub-periods. The results support evidence of time-varying dependence among the crude oil, U.S., and South Korean markets. These results have important implications for the analysis of port- folio risk management and for assisting energy policymakers and traders in making effective decisions and investments, respectively.” 

#Comment 2. While authors respond to these comments, readers would prefer that the adding context or revised context would be shown right below the response to comments, since readers might not prefer the presentation as shown below. 

In addition, the same concerns for responding to other comments. 

-> Thank you for your valuable comment. As requested, we put the adding context below the response to the comment. 

We additionally provide the previous “Response to Reviewers” below. We appreciate your comments.

—————————————————————————————————————————————————————————————————————————————————————

#Comment 2. Authors mention that “observing oil price volatility and taking actions in response to its expected changes are essential for managing risk” would be important for the relevant literature. We argue that the above concern might be already taken into account in the relevant studies. As a result, we document that authors have to mention the main differences between your study and other relevant studies in detail. In addition, how to manage the risks in terms of oil price fluctuated considerably should be illustrated in detail as well.

-> Thank you for your valuable comment. As requested, we mentioned the main differences between our study and other relevant studies in detail. The following sentence are added to the Introduction, Section Discussion in terms of shale gas and risk management and Section Concluding remarks:

Introduction section, Page 3, lines 79 - 86 :

”There are two aspects of this study that differ from previous studies. The first is our use of volatility indices to identify the relationship between crude oil and the stock market. Although previous empirical studies find causal relationships between oil prices and stock indices, research on the causality between implied volatility indices is scarce. To bridge this gap, we adopt the OVX, VIX, and VKOSPI to measure the implied volatility in oil prices, the S&P 500, and the KOSPI 200, respectively. Second, whereas previous studies focus mainly on the relationship between crude oil and the stock market, we focus on the change in that relationship over time.” 

Discussion in terms of shale gas and risk management section, Page 19 - 20, lines 472 - 485 : 

“The difference between the existing studies and this study is that we use volatility indices to identify the relationship between crude oil and the stock market. The volatility index represents a measure of the risk and it implies the market participants’ expectations for the market. Therefore, we can see whether the expected risk predicted by each market participant affects the expectations of other market participants by looking at the changes in the relationship between the volatility indices. Our empirical results indicate that forecasts of risks to crude oil and risks to the stock market were mutually influenced prior to 2014, but recently the effects have been reduced. Furthermore, we explain the relationships between the U.S. and South Korea stock markets, and between crude oil and the South Korean stock market with respect to the volatility indices. The studies that examine the relationship between crude oil and U.S. stock markets are quoted in Section 2. There are also studies on the effect of changes in crude oil prices on the South Korean economy(Masih et al. [2011],Ran and Voon [2012],Wang et al. [2013]), but they use crude oil prices and the South Korean stock market index not volatility indices. Like- wise, Jeon and Jang [2004] and Kim and Ryu [2015] examine the relationship between stock prices in the U.S. and South Korean stock markets, but they use stock indices.” 

Concluding remarks section, Page 21, lines 536 - 539 : 

”This study is noteworthy in that the influence of crude oil volatility on the U.S. and South Korean stock markets has decreased significantly. In addition, it is remarkable that the influence of the U.S. stock market on the South Korean stock market has increased.” 

Furthermore, we added a section(Section Discussion in terms of shale gas and risk management) to state how to manage the risks in terms of oil price fluctuated in detail. In the section, we illustrated a way to manage the risk of oil price fluctuation for a portfolio of oil and stocks, according to the relevant literature. We mentioned the calculation of optimal portfolio weights and hedge ratios as a future study. The following sentence are added: 

Discussion in terms of shale gas and risk management section, Page 20, lines 487 – 493: 

”We can manage the volatility risk arising from the crude oil price fluctuations by calculating optimal portfolio weights and hedge ratios. Assume that an investor attempts to hedge exposure to crude oil price fluctuation for a portfolio of oil and stocks. Then, the investor wants to minimize the risk of his/her oil-stock portfolio without reducing its expected returns. According to Kroner and Ng [1998], conditional volatilities can be used to construct optimal portfolio weights … between oil and stock returns at time t, respectively. Therefore, when we calculate the conditional volatilities by using some models, such as the GARCH and BEKK models, we obtain a dynamic hedge ratio. Furthermore, given an OVX derivatives and stocks portfolio, we can also apply this process to calculate the optimal hedge ratio.” 

Concluding remarks section, Page 21, lines 558 – 559: 

”Possible future studies include research on optimal portfolio weights and hedge ratios with respect to the sub-period data used in this study.” 

#Comment 3. The motivation of this study should be enhanced and appeared in the front of this paper since readers prefer the strong motivation presented in the beginning of the introduction section. 

-> Thank you for your valuable comment. As requested, we enhanced the motivation of this study and presented it in the Introduction section. The following sentence are added: 

Introduction section, Page 2, lines 42 – 51: 

”Most research still uses crude oil and stock prices. However, the volatility indices are a better suitable barometer of the fragility of the markets and the economy. Therefore, the aim of this work is to investigate the relationship among the volatility indices, to derive important implications for the analysis of portfolio risk management. Furthermore, since the introduction of volatility derivatives (e.g., Chicago Board Options Exchange (CBOE) volatility index (VIX) futures, options, and exchange-traded products), the trading volume has been increasing because they can be used as a risk-hedging strategy against stock market downturns (e.g. Park [2016]). Accordingly, investigation of the relationship between volatility indices can give necessary insight into suggestions for the pricing of volatility derivatives.” 

#Comment 4. Authors had better provide the reasons why authors adopt these methodologies (i.e. the autoregressive distributed lag (ARDL) bounds test of cointegration as well as the TodaYamamoto (TY) Granger causality test of Toda and Yamamoto (1995)). Why not employing other methodologies due to one of the methodologies proposed in 1995? Are there any other update and appropriate methodologies likely employed for this paper? 

-> Thank you for your valuable comment. As requested, we introduced the cointegration test and mentioned the reason why we adopt the ARDL bounds test. About the ARDL bounds test, the following sentence are added: 

ARDL bounds tests subsection, Page 9 – 10, lines 199 – 208 :

”The cointegration tests proposed by Engle and Granger [1987], Johansen [1991], and Johansen and Juselius [1990] have been used in many empirical studies to investigate the long-run relationship of economic variables. However, the use of these approaches is limited. For example, these methods can be applied to those series that have a unique order of integration. The ARDL bounds test proposed by Pesaran and Shin [1998] and Pesaran et al. [2001] is a popular method because it has certain advantages over traditional cointegration methods. 

First, it does not need all the variables in the model to be integrated of the same order. Second, the approach is relatively more efficient in the case of small and finite sample data sizes (Pesaran and Shin [1998] and Tang [2002]). Third, applying the ARDL technique, we obtain unbiased estimates of the long-term model (Harris and Sollis [2003]).” 

About TY Granger causality test, we introduced the causality test and mentioned the reason why we adopt the TY Granger causality test. Furthermore, we cited some studies in which TY causality test is used. We emphasize the following sentence: 

TY Granger causality tests subsection, Page 11, lines 236 – 247: 

”Granger [1969] proposes a test of the causal relationship between two variables, known as Granger causality. A time series (X) is said to Granger cause another time series (Y ) if the prediction error of the current Y declines by using the past values of X in addition to the past values of Y . In the test, the two variables are expressed by simple vector autoregression (VAR). The Granger causality test is easy to implement and can be applied in many types of empirical studies. Nonetheless, it also has some drawbacks. According to Shirazi and Manap [2005], first, the Granger causality test for inferring the leads and lags among integrated variables can provide spurious regression results. Second, it does not consider the effect of the number of lags even though this can affect the performance of a causality test. In other words, the results of the Granger causality test depend on the number of lags. Moreover, Toda and Phillips [1993] insist that Granger causality can lead to drawing wrong conclusions because of the dependence of the parameters.” 

TY Granger causality tests subsection, Page 11, lines 264 – 274: 

”The TY Granger causality test has several advantages over other methods. First, it can provide a valid result regardless of whether a series is I(0),I(1), or I(2), not cointe- grated, or cointegrated of any arbitrary order. Second, the TY test avoids the bias associated with unit root and cointegration tests (Zapata and Rambaldi [1997],Clarke and Mirza [2006]) as it does not require pre-testing of the cointegrating properties of the system. Third, we can explore the causality between variables with a possibly integrated and cointegrated system using the augmented VAR model in the TY test because the long-run information of the system in the general VAR model often dis- appears in the mandatory process of first differencing and pre-whitening (Clarke and Mirza [2006],Jain and Ghosh [2013]). Therefore, we adopt TY causality testing in this study. Furthermore, many recent studies adopt the approach of identifying causal- ity using the TY causality test (Adriana [2014],Ghosh and Kanjilal [2014],Ziramba [2015],Faisal et al. [2017],Sankaran et al. [2019]).” 

In addition, we referred to developed methodologies for Causality test since TY Granger causality test of Toda and Yamamoto (1995)). They are nonlinear version, asymmetric, and bootstrap panel causality tests. We mentioned the nonlinear causality tests as a future study. The following sentence are added: 

TY Granger causality tests subsection, Page 11, lines 254 – 263: 

”Several other methodologies have been developed since the TY causality test was introduced. In view of the TY causality test being limited to finding linear cause-effect relation- ship, nonlinear version Granger causality tests have been developed(e.g. Hiemstra and Jones [1994] and Kyrtsou and Labys [2006]). We can detect nonlinear interac- tions between the variables using these tests. By contrast, Hatemi-j [2012] proposes an asymmetric causality test for the existence and direction of causality. Thus, the causality between positive and negative shocks of variables can be determined in the method. In addition, Ko ´nya [2006] introduces a bootstrap panel causality method in order to account for both cross-sectional dependence and slope heterogeneity(Kar et al. [2011]). The approach is widely used to test for causality in a panel framework in many empirical studies.” 

Concluding remarks section, Page 21, lines 559 – 562: 

”In addition, we can consider the causality between the positive and negative shocks of volatility indices by using nonlinear causality tests. In other words, it is necessary to ascertain how they affect and receive each other when market risk increases and when it decreases.” 

#Comment 5. In fact, ARCH effects are likely existed in financial time series. We suggest that authors have better employ Multivariate GARCH-family models to examine the relationships among the variables (i.e. OVX, VIX, and VKOSPI) employed in this study for robustness. 

-> Thank you for your valuable comment. As requested, we employ BEKK-GARCH model as multivariate GARCH-family model to examine the relationships among volatility indices. We briefly introduced BEKK-GARCH model in subsection Multivariate GARCH model and added empirical results by the BEKK-GARCH model were given in subsection BEKK- GARCH(1,1) model and section Sub-period analysis. These results are also consistent with the causality test results obtained by TY Granger causality test. 

#Comment 6. The contribution of this study could be enhanced since the presentation of the contribution of this study might not appeal readers. 

-> Thank you for your valuable comment. As requested, we enhanced the contribution of this study and mentioned three main contributions in the Introduction. The following sentence are added: 

Introduction section, Page 3, lines 87 – 103: 

”We obtain three main contributions from these differences. The first is the investigation of the relationship between future expectations for each market—the crude oil, U.S., and South Korean stock markets. In particular, the volatility index represents the future risk measure of market participants. Therefore, we can investigate the relationship between the risk measures implied by crude oil, the S&P 500 index, and the KOSPI 200 index by using the volatility indices. The second is the examination of the causality be- tween the OVX and VKOSPI and between the VIX and VKOSPI. To the best of our knowledge, this study is the first to investigate the relationship between the OVX and VKOSPI. Based on their relationship, policymakers can propose laws and policies for oil-importing countries to manage market risk. As mentioned above, South Korea and the United States have a close economic relationship; hence, it is reasonable to explore the causality between them owing to the uncertainty in their stock markets. The third major contribution concerns the change in the relationship between the volatility indices as revealed through a sub-period analysis. Based on the empirical results of the sub-period analysis, we conclude that one of the factors causing the change in the relationship is the increased production of shale gas. Detailed discussions on this will be covered in Section 6.” 

In order to appeal readers, we presented some applications and implications based on this empirical results. The following paragraph is added to the Section Concluding remarks: 

Concluding remarks section, Page 21, lines 540 – 557: 

”We present some applications and implications based on this study’s results. First, for both investors and policymakers, the key application of our work is properly forecasting financial market volatility. In particular, changes in the VIX in the U.S. stock market are strongly related to those in the South Korean stock market. In other words, we can increase the predictive power of the future VKOSPI in the South Korean stock market using the movement of the VIX. As Gong and Lin [2018] claim, forecasting volatility indices may be more beneficial to the decision-making of all stock market participants (including financial traders, manufacturers, and policymakers). Second, because volatility indices are used to hedge volatility risk, our findings will help to manage volatility risk in crude oil portfolios. According to Chen et al. [2011] and Liu et al. [2013], OVX derivatives, futures, and options can be a financial tool to hedge volatility risk. Furthermore, volatility indices have become a popular asset class for investors considering diversifying their portfolio strategy. Thus, our empirical findings can be used to examine and evaluate volatility derivatives, such as OVX options and futures. Third, contrary to expectations, South Korea, an emerging market, has not been sensitive to crude oil risks lately. There are many reasons for this, but oil import diversification may be one of them. Therefore, this study can be seen as evidence of the effect of crude oil import diversification for oil-importers, and in particular, South Korea.” 

#Comment 7. Due to the data period employing in this study from January 2, 2009 to December 28, 2018, we argue that authors had better to examine whether there are any structural changes existed over the data period instead of examining the relationship among these variables by using two almost equal sub-periods (i.e. two sub-periods, namely January 2009May 2014 (sub-period 1) and June 2014December 2018 (sub-period 2). 

-> As requested, we examined the structural changes over the data period and calculated structural breakpoints for OVX time series by the algorithm described in [Bai and Perron, 2003]. In addition, we referred to some studies in which the sub-period analysis was investigated. The following sentence are added: 

Sub-period analysis section, Page 16, lines 383 – 390: 

”In addition to investigating the relationship between the volatility indices for the entire period (2009– 2018), the total sample is examined for structural breaks in OVX by using the Bai and Perron [2003] sequential breakpoint tests. According to the breakpoint tests, the entire sample is split into two sub-periods after locating the date of 10/8/2014 as the breakpoint. Therefore, we analyze two sub-periods, namely January 2, 2009–October 7, 2014 (sub-period 1), and October 8, 2014–December 28, 2018 (sub-period 2). The OVX time series for the two sub-periods are illustrated in Fig 2 with the breakpoint. This sub-period analysis is often carried out in other studies(Nazlioglu et al. [2013], Guesmi and Fattoum [2014], Kayalar et al. [2017], Pavlova et al. [2018]).”

---

## [Decision Letter · Decision Letter 3]

17 Apr 2020

Relationship between uncertainty in the oil and stock markets before and after the shale gas revolution: Evidence from the OVX, VIX, and VKOSPI volatility indices

PONE-D-19-34694R3

Dear Dr. choi,

We are pleased to inform you that your manuscript has been judged scientifically suitable for publication and will be formally accepted for publication once it complies with all outstanding technical requirements, respectively abstract revision.

With kind regards,

Stefan Cristian Gherghina, PhD. Habil.

Academic Editor

PLOS ONE

Additional Editor Comments (optional):

Reviewers' comments:

Reviewer's Responses to Questions

**Comments to the Author**

1. If the authors have adequately addressed your comments raised in a previous round of review and you feel that this manuscript is now acceptable for publication, you may indicate that here to bypass the “Comments to the Author” section, enter your conflict of interest statement in the “Confidential to Editor” section, and submit your "Accept" recommendation.

Reviewer #1: All comments have been addressed

Reviewer #2: All comments have been addressed

2. Is the manuscript technically sound, and do the data support the conclusions?

Reviewer #1: Yes

Reviewer #2: Yes

3. Has the statistical analysis been performed appropriately and rigorously? 

Reviewer #1: Yes

Reviewer #2: Yes

4. Have the authors made all data underlying the findings in their manuscript fully available?

Reviewer #1: Yes

Reviewer #2: Yes

5. Is the manuscript presented in an intelligible fashion and written in standard English?

Reviewer #1: Yes

Reviewer #2: Yes

6. Review Comments to the Author

Reviewer #1: 1. In fact, the abstract should be neat and focused on your main concern in this paper, so we suggest that the abstract should be condensed in accordance to the above-mentioned concern.

Decision: minor revision

Reviewer #2: No further comment. Authors has addressed the issues mentioned in the report. I believe the overall quality of the paper has met the journal requirements.

7. PLOS authors have the option to publish the peer review history of their article (what does this mean?). If published, this will include your full peer review and any attached files.

Reviewer #1: No

Reviewer #2: No

---

## [Editor Report · Acceptance letter]

21 Apr 2020

PONE-D-19-34694R3 

Relationship between uncertainty in the oil and stock markets before and after the shale gas revolution: Evidence from the OVX, VIX, and VKOSPI volatility indices 

Dear Dr. Choi:

I am pleased to inform you that your manuscript has been deemed suitable for publication in PLOS ONE. Congratulations! Your manuscript is now with our production department. 

With kind regards,

on behalf of

Dr. Stefan Cristian Gherghina 

Academic Editor

PLOS ONE